# Conformational changes in twitchin kinase in vivo revealed by FRET imaging of freely moving *C. elegans*

**Daniel Porto[1], Yohei Matsunaga[2], Barbara Franke[3], Rhys M Williams[3], Hiroshi Qadota[2], Olga Mayans[3], Guy M Benian[2], Hang Lu[1,4]***

[1]Interdisciplinary Bioengineering Program, Georgia Institute of Technology, Atlanta, United States; [2]Department of Pathology, Emory University, Atlanta, United States; [3]Department of Biology, University of Konstanz, Konstanz, Germany; [4]School of Chemical & Biomolecular Engineering, Georgia Institute of Technology, Atlanta, United States

**Abstract** The force-induced unfolding and refolding of proteins is speculated to be a key mechanism in the sensing and transduction of mechanical signals in the living cell. Yet, little evidence has been gathered for its existence in vivo. Prominently, stretch-induced unfolding is postulated to be the activation mechanism of the twitchin/titin family of autoinhibited sarcomeric kinases linked to the mechanical stress response of muscle. To test the occurrence of mechanical kinase activation in living working muscle, we generated transgenic *Caenorhabditis elegans* expressing twitchin containing FRET moieties flanking the kinase domain and developed a quantitative technique for extracting FRET signals in freely moving *C. elegans*, using tracking and simultaneous imaging of animals in three channels (donor fluorescence, acceptor fluorescence, and transmitted light). Computer vision algorithms were used to extract fluorescence signals and muscle contraction states in each frame, in order to obtain fluorescence and body curvature measurements with spatial and temporal precision in vivo. The data revealed statistically significant periodic changes in FRET signals during muscle activity, consistent with a periodic change in the conformation of twitchin kinase. We conclude that stretch-unfolding of twitchin kinase occurs in the active muscle, whereby mechanical activity titrates the signaling pathway of this cytoskeletal kinase. We anticipate that the methods we have developed here could be applied to obtaining in vivo evidence for force-induced conformational changes or elastic behavior of other proteins not only in *C. elegans* but in other animals in which there is optical transparency (e.g., zebrafish).

*For correspondence: hang.lu@gatech.edu

Competing interest: The authors declare that no competing interests exist.

## Introduction

Cells and tissues invariably require mechanical stimulation for development, differentiation, and physiological maintenance. While the role of proteins as chemosensors in cellular signaling cascades is well established, the molecular mechanisms underlying the sensing and transduction of mechanical signals remain poorly understood. Accumulating evidence suggests that proteins can act as mechanotransductors by undergoing repeated cycles of unfolding and refolding when subjected to directional forces (*Hu et al., 2017*; *Sharma et al., 2020*). This mechanism appears to apply in particular to multidomain proteins that can bear the unfolding of mechanolabile segments of their chains, while maintaining the structural integrity of their other functional regions (*Sharma et al., 2020*). The feasibility of such stretch-unfolding in proteins is evidenced by numerous in vitro and computational studies that use atomic force microscopy (AFM), optical tweezers, and steered molecular dynamics simulations (SMDS) (reviewed in *Batchelor et al., 2020*). As a result, force-induced unfolding is now emerging as

a possible general and central mechanism in mechanical sensing. However, little evidence of the existence of this mechanism in vivo exists to date and, thus, its physiological relevance remains strongly debated.

A paradigm of a biological mechanotransductor is the muscle sarcomere, whose trophicity is regulated by mechanical demand. Mounting evidence indicates that the giant proteins of the twitchin/titin family (0.7–4 MDa) are key force-sensing molecules in the sarcomere (e.g. *Butler and Siegman, 2011*; *Buyandelger et al., 2014*). These proteins form elastic intrasarcomeric filaments and contain one or two kinase domains, which are thought to mediate the response of muscle to mechanical stress (*Kontrogianni-Konstantopoulos et al., 2009*). Nematode twitchin kinase (TwcK) is the best studied member of the family and is the closest homolog to human titin kinase (TK). TwcK 3D-structure shows that the kinase domain is enwrapped by N- and C-terminal tail extensions (termed NL and CRD) that bind against the kinase hinge region and active site, respectively, leading to a complete inhibition of its phosphotransfer activity (*Hu et al., 1994*; *Lei et al., 1994*; *von Castelmur et al., 2012*). The truncation of the NL and/or CRD extensions has been shown to result in active TwcK (*von Castelmur et al., 2012*). Human TK shares its 3D-architecture with TwcK (*Mayans et al., 1998*; *Bogomolovas et al., 2021*). AFM data and SMDS have led to the proposal that the regulatory tails of TwcK and TK are extensible, becoming dynamically and reversibly unfolded by the pulling forces that develop in the sarcomere during muscle contractile activity (*von Castelmur et al., 2012*; *Gräter et al., 2005*; *Puchner et al., 2008*; *Puchner and Gaub, 2010*). The stretch-unfolding of TwcK flanking tails would expose the catalytic center in the kinase and, thereby, lead to its phosphotransfer activation. The view that stretch activates TwcK has been supported in *Mytilus*; using permeabilized smooth muscles, a 10 % stretch resulted in a twofold increase in phosphorylation of a model substrate for molluscan TwcK in vitro (*Butler and Siegman, 2011*). For *Caenorhabditis elegans*, TwcK phosphotransfer catalysis has been revealed to regulate muscle contraction (*Matsunaga et al., 2017*) and TwcK has been shown to be the target of the MAK-1/p38 MAPK pathway that is linked to cellular stress, so that TwcK is hypothesized to be the avenue by which stress factors affect muscle mechanics in *C. elegans* (*Matsunaga et al., 2015*). Human TK has been shown to be associated with the autophagosomal receptors nbr1 and p62 and the E3 ubiquitin ligases MuRF1 and MuRF2 and, thereby, it has been linked to turn-over and gene expression patterns in the myofibril in function of mechanical activity (*Lange et al., 2005*; *Bogomolovas et al., 2014*; *Bogomolovas et al., 2021*). Hence, the mechano-sensing properties of these sarcomeric kinases are speculated to link the proteostatic response of the myofibril to the mechanical demand imposed on the sarcomere.

Despite the significance of the hypothesized stretch-induced unfolding mechanism in the sarcomere, the existence of this process in muscle in vivo remains untested. To address this question, we have created transgenic *C. elegans* carrying FRET moieties that can report on the conformational change occurring in the kinase region of twitchin. We further developed a custom tracking platform and an analysis protocol to quantitatively measure in vivo FRET signals in working muscles in freely moving animals. To our knowledge, this is the first demonstration of spatiotemporal quantification of FRET within the whole body of freely moving *C. elegans*. The tracking platform employed three-color imaging with high spatial and temporal precision, along with image processing algorithms to accurately extract fluorescence and posture measurements and remove artifacts derived from the free motion of the imaged animals. We demonstrate that as the worm moves, there is a detectable and statistically significant change in FRET signal that correlates with the contraction-relaxation mechanical cycle of muscle. The FRET signal is maximal during muscle contraction and minimal during muscle stretch. This suggests that physiological forces acting on twitchin kinase in its native in vivo context are capable of inducing stretch-unfolding events and, thus, that this is a physiologically relevant mechanism in live muscle.

## Results

### Design and validation of twitchin kinase FRET chimeras

The kinase region of the twitchin protein from *C. elegans* is a multi-domain segment of composition Fn-NL-kinase-CRD-Ig, where Fn is a fibronectin domain, NL is a 45-residue sequence, CRD is a 60-residue sequence, and Ig is an immunoglobulin domain (*Figure 1A*). NL and CRD sequences bind onto the catalytic kinase domain, inhibiting its catalysis (*von Castelmur et al., 2012*). In order to

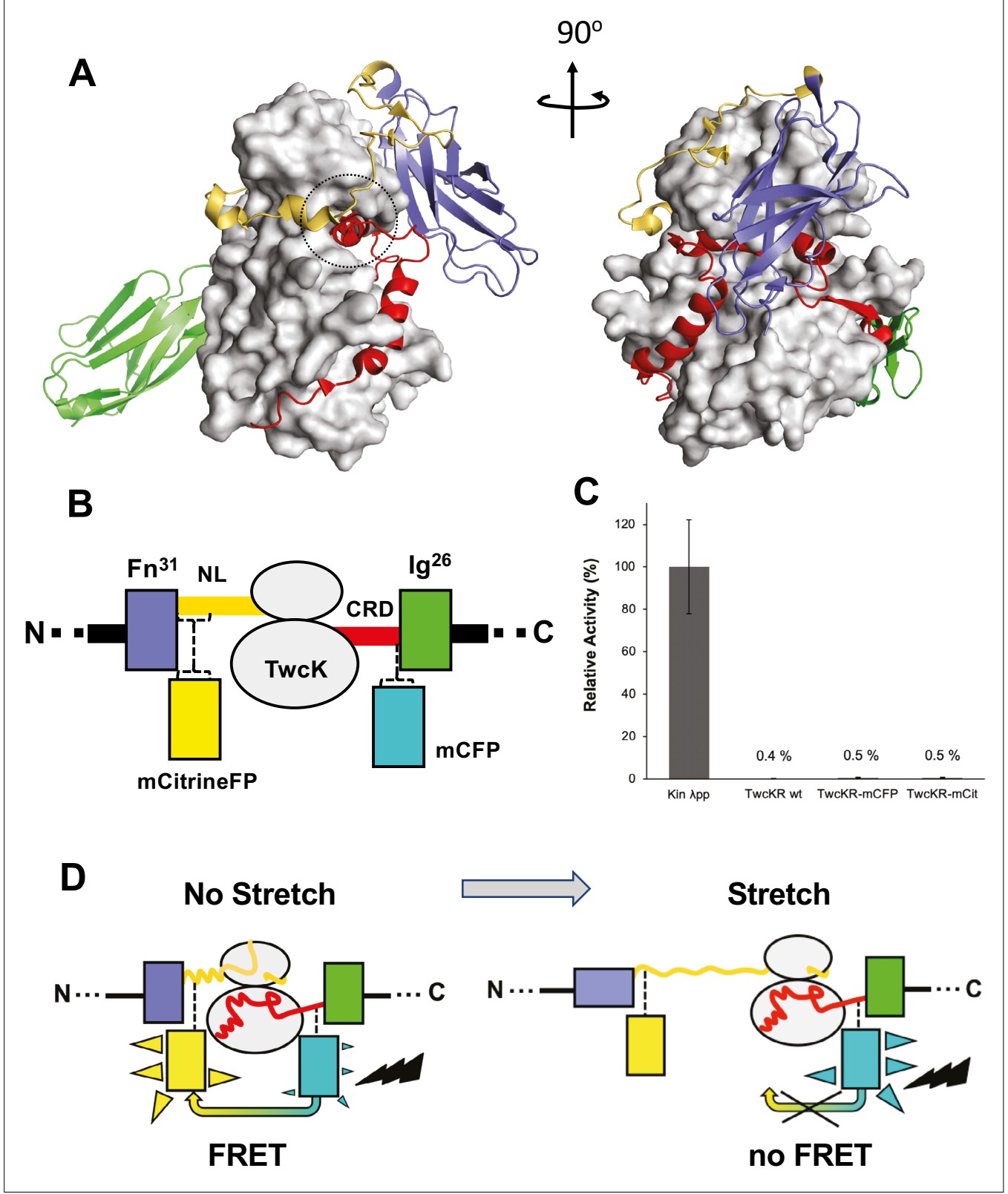

**Figure 1.** Molecular composition of the TwcK-FRET sensor. (**A**) Crystal structure of the multi-domain, autoinhibited TwcK (PDB code 3UTO). The ATP pocket, blocked by the CRD extension, is marked with a circle. Domain color code as described in (**B**). (**B**) Domain composition of the TwcK region, illustrating the sites of mCitrineFP and mCFP incorporation. (**C**) Activity measurements of multi-domain TwcK variants using the Kinase Glo luminescent assay (Kin $\lambda$ pp) indicate the twitchin kinase domain that displays maximal activity by lacking flanking tails and having been treated with $\lambda$ phosphatase

*Figure 1 continued on next page*

*Figure 1 continued*

(*Williams et al., 2018*). (**D**) Functional principle of the TwcK-FRET stretch-sensing construction. The non-stretched state represents the basal molecular state, where NL and CRD tail extensions pack against the kinase. Unfolding and extension of the NL segment (and/or CRD sequence) are expected to happen concurrently and result in a stretched state.

monitor the unfolding of NL and/or CRD sequences induced by sarcomeric stretch and, thereby, the freeing of twitchin kinase, we constructed a multi-domain kinase chimera carrying fluorescent proteins (FPs) inserted N- and C-terminally to the NL-Kin-CRD assembly, to act as a FRET stretch sensor in vivo. The resulting chimera had the domain composition Fn-mCitrineFP-NL-TwcK-CRD-mCyanFP-Ig (abbreviated in the following as TwcK-FRET; *Figure 1B*). For the design of the TwcK-FRET chimera, the crystal structure of the multi-domain TwcK region (*Figure 1A*) (Protein Data Bank entry 3UTO; *von Castelmur et al., 2012*) was first examined visually (PyMOL; *DeLano, 2002*) and regions deemed not critical for structural integrity identified. These regions would then accommodate FP moieties. This evaluation considered the location of each residue with respect to mechanistically important loci in TwcK (*von Castelmur et al., 2012*; *Williams et al., 2018*) as well as the number and type of intramolecular contacts sustained by each residue. Importantly, locations selected for the insertion of FPs had to be proximal (<7–8 nm), as to allow for robust FRET to occur (*Evers et al., 2006*; *Ohashi et al., 2007*). Applying these criteria, dispensable structural elements—whose alteration would not disrupt the packing of the inhibitory NL and CRD tails against the kinase domain or affect the integrity of the kinase active site—were identified. Specifically, two loci were selected: a flexible seven-residue segment in the frontal fraction of the NL sequence (6207-DERKRRR-6213); and a two-residue motif (6584-QP-6585) in the linker sequence between CRD and domain Ig26. These sites were located at a respective distance of ~5.3 nm (roughly coincident with the 4.8 nm Förster distance of the introduced FP pair *Lam et al., 2012*).

The sites so identified were modified to include mCitrine and mCyan FP FRET reporters, respectively. Both FPs are enhanced variants of the green FP from *Aequorea victoria* (*Griesbeck et al., 2001*; *Zacharias et al., 2002*). Important for the present work is that both FP variants are monomeric (*Shaner et al., 2005*), which minimizes the risk of undesirable mutual interactions within the sarcomeric lattice. mCyan FP (mCFP) is a cyan FP that acts as FRET donor and has $\lambda_{Ex}/\lambda_{Em}$ maxima at 433/475 nm. The mCitrine FP is an enhanced yellow FP that acts as FRET acceptor, having $\lambda_{Ex}/\lambda_{Em}$ maxima at 516/529 nm. For the inclusion of mCitrine FP, the flexible loop sequence 6207-DERKRRR-6213 (~1.6 nm end-to-end distance) was excised and replaced with the FP, including a GSG linking motif at each terminus. The mCFP sequence and flanking GSG residues were inserted between the two residues in the 6584-QP-6585 locus. This resulted in the Fn-mCitrineFP-NL-Kin-CRD-mCFP-Ig protein chimera (*Figure 1B*). In this construction, the compact slack form of the kinase (where the NL and CRD tails pack onto the kinase domain) will yield maximal FRET signal, while the stretch-induced unraveling of NL or CRD sequences will result in a decrease or loss of FRET (*Figure 1D*).

To test whether the inserted FP proteins had any unwanted effects on the structural integrity of TwcK, the catalytic activity of recombinantly expressed TwcK-mCFP (Fn-NL-Kin-CRD-mCFP-Ig) and TwcK-mCitrineFP (Fn-mCitrineFP-NL-Kin-CRD-Ig) proteins was measured in a phosphotransfer assay. The NL and CRD regulatory extensions cooperatively suppress catalytic activity in TwcK while the maximal TwcK activity is displayed by the kinase domain lacking both NL and CRD tails (*von Castelmur et al., 2012*). The disruption of either tail leads to partial, but significant, levels of catalysis (*von Castelmur et al., 2012*). Catalytically active TwcK-FRET constructs would therefore be indicative of a disruption of the packing of the autoinhibitory tails and, thereby, of a compromised structural integrity of the NL-Kinase-CRD assembly. In this study, catalytic assays showed that both TwcK-mCFP and TwcK-mCitrineFP samples have negligible activity, equivalent to that of wild-type multi-domain TwcK (*ca* 0.5 % of the maximal activity of the free kinase domain) (*Figure 1C*). This proved that the insertion of FP proteins flanking TwcK for the purpose of engineering a FRET sensor had not caused an undesirable perturbation of the native structure of the multidomain kinase.

## Generation of transgenic *C. elegans* expressing twitchin kinase FRET chimeras

To monitor the FRET signal of the modified TwcK in vivo, *C. elegans* strains were generated using two different strategies. First, we created transgenic strains expressing multi-domain TwcK-FRET segments

similar to those designed and validated biochemically (*Figure 1*). Because transgenic methodology in *C. elegans* involves the use of concatemers of DNA segments containing many copies of a gene, these transgenic strains express more than the normal levels of a protein as compared to expression from an endogenous single-copy gene. However, we took advantage of this overexpression to obtain FRET signals of high intensity from the TwcK-FRET segments, allowing us to develop our methodology for tracking, recording, and analyzing FRET in freely moving *C. elegans*. Once the methodology had been established, and to overcome the potential risk of the overexpressed fragments not mimicking the behavior of the twitchin protein in its native sarcomere context, we used CRISPR/Cas9 to edit the twitchin gene to express full-length twitchin containing mCitrineFP and mCFP moieties positioned as in the tested fragments, and expressed at normal levels from the endogenous twitchin gene.

Thus, initially, *C. elegans* strains in our study expressed various portions of twitchin, all comprising the multi-domain kinase segment, under the control of a muscle-specific promoter and with an HA tag. Endogenous twitchin localizes to sarcomeric A-bands in a characteristic manner in which there is a lack of localization in the middle of A-bands (*Moerman et al., 1988*). Using anti-HA antibodies to visualize the fragments (*Figure 2A*), we identified a minimal twitchin fragment that displayed normal twitchin A-band localization: Ig-Ig-Fn-NL-Kin-CRD-Ig-Ig-Ig-Ig-Ig. We next inserted mCitrineFP and mCFP in this minimally localizing fragment in the loci described. The resulting fragment, Ig-Ig-Fn-mCitrineFP-NL-Kin-CRD-mCFP-Ig-Ig-Ig-Ig-Ig, also localized normally to A-bands when expressed in transgenic worms (strain GB282) (*Figure 2B*). As a 'negative control,' we created a transgenic animal expressing the same minimally localizing fragment but carrying FRET moieties placed on either side of an Ig domain C-terminal to the kinase domain, Ig28 (strain GB284) (*Figure 2C*). However, AFM measurements have confirmed that Ig28 has a mechanically stable fold and that it will undergo stretch-induced unfolding only under extreme force (*Greene et al., 2008*), making this a non-extensible construction so that its FRET moieties are not expected to alter significantly their mutual distance at physiological muscle force. Ig28 is a type-I immunoglobulin domain joined serially to its flanking domains by short linker sequences, so that the distance between the FRET loci in this variant is 4.5–5 nm. Hence, the distances between FRET moieties in the control and test constructs in this study are comparable (*Figure 2D*). The control sample Ig-Ig-Fn-NL-Kin-CRD-Ig-Ig-mCFP-Ig-mCitrineFP-Ig-Ig, when expressed in transgenic worms, also localized correctly to A-bands (*Figure 2C*). A similar localization of test and control fragments was expected as the twitchin segment is fundamentally the same in both cases: namely, test and control constructs share the same twitchin backbone, have the same chemical composition and carry the same FRET moieties at comparable distances. To quantitate the expression of test and control twitchin segments in strains GB282 and GB284, respectively, we performed quantitative western blots using total protein extracts and anti-HA antibodies. As shown in *Figure 2—figure supplement 1*, the control strain GB284 expressed twofold higher FP-labeled twitchin segments than the tester strain GB282 (with N=4, mean=2.10). Importantly, as our study is based on ratiometric FRET differences, the divergence in protein expression levels would not affect the calculated signal directly. Only the FP labels in the test construct would be expected to alter their FRET signal significantly during muscle function as the NL and the CRD are the only extensible segments in this twitchin fragment, and thus, our expectation was that ΔFRET values would be higher for the test than the control strain independently of protein expression levels.

## Precision measurements of FRET signal and curvature in freely moving animals

To identify the relationship between twitchin kinase conformation and sarcomere mechanics, we sought to record fluorescence signals in vivo in freely moving animals. Muscle contraction cannot be directly measured with precision in moving *C. elegans* because of the size of the worm and the resolution of microscopy techniques. *C. elegans* move in an undulatory motion most of the time (with a periodicity of ~1 Hz) and its body wall muscle lies directly under the cuticle and is attached to the cuticle via a basement membrane and a thin hypodermal or skin cell layer. Thus, we reasoned that local curvature is a good proxy for the mechanical state of the sarcomere. We defined positive curvature values to represent contracted muscles, and negative curvature values to represent relaxed and stretched muscles. We designed an approach to simultaneously measure the FRET signal of the twitchin chimeras in vivo and the local curvature of the sarcomere where the FRET is measured.

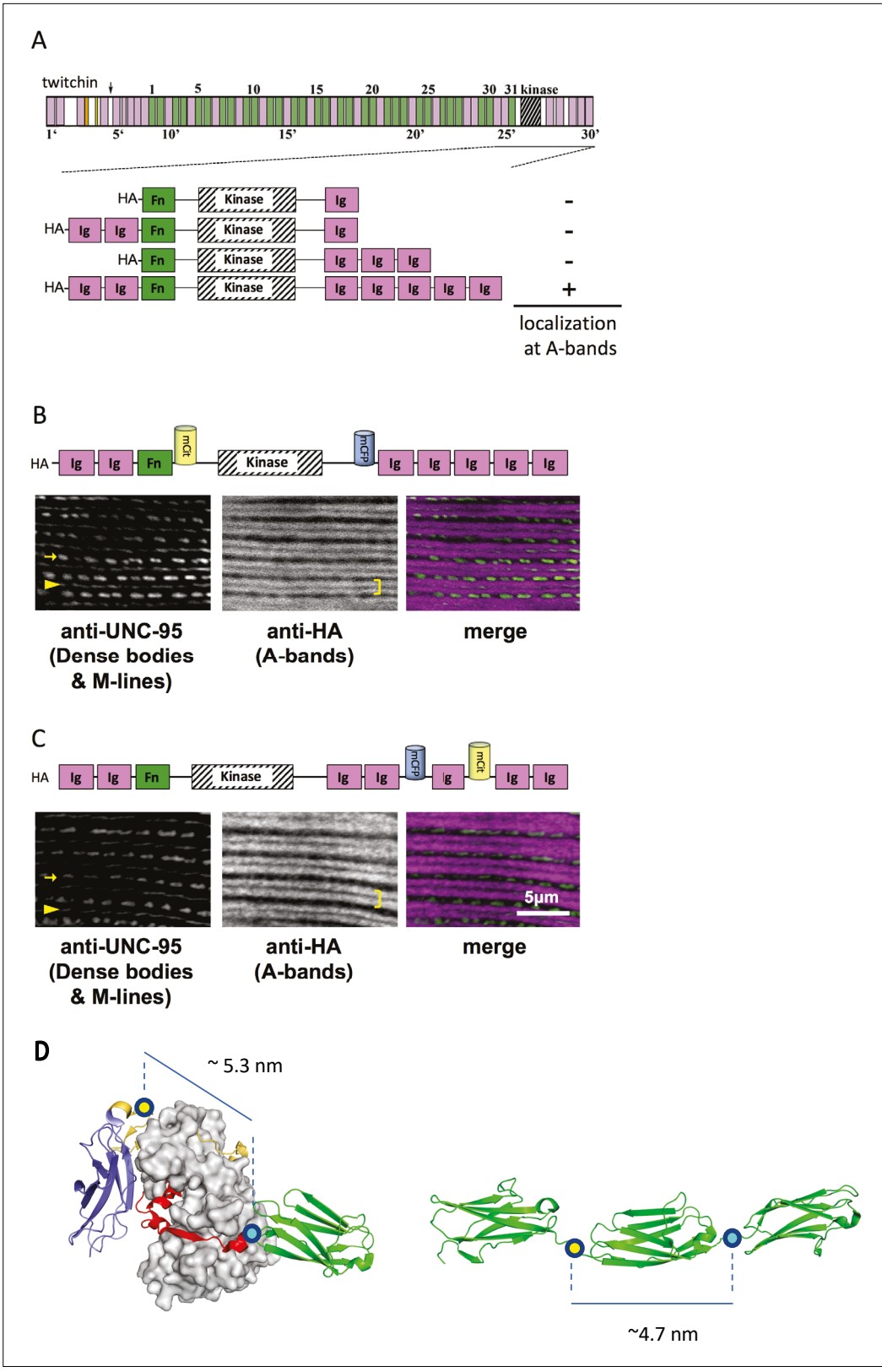

**Figure 2.** A portion of twitchin containing FRET moieties properly localizes in the sarcomere. (**A**) Schematic representation of domain organization of twitchin and segments containing the kinase domain tested for localization to A-bands. Note that HA-Ig-Ig-Fn-NL-Kin-CRD-Ig-Ig-Ig-Ig-Ig localizes to sarcomeric A-bands, the normal location of full-length twitchin. (**B**) The minimally localizing twitchin fragment containing FRET moieties

*Figure 2 continued on next page*

*Figure 2 continued*

surrounding the kinase domain localizes to A-bands. Above, schematic; below, immunofluorescent localization. (**C**) The minimally localizing twtichin fragment containing FRET moieties surrounding Ig28 also localizes to A-bands. Above, schematic; below, immunofluorescent localization. Arrow, row of dense bodies; arrowhead, M-line; bracket, A-band. (**D**) Structural representations of the test (left) and control (right) molecular loci, including estimated distances between the sites of FP incorporation. The test molecule is the crystal structure of the multi-domain twitchin kinase (PDB 3UTO). The control molecule is a model of the Ig27-Ig28-Ig29 segment based on available structural data for homologous poly-Ig tandems of human titin (**von Castelmur et al., 2008**).

The online version of this article includes the following figure supplement(s) for figure 2:

**Figure supplement 1.** As controls for artifacts, analysis on individual fluorophores showing the absence of consistent increases in change and correlations in all parts of the contraction cycle, comparing GB282 to GB284.

To measure both curvature and FRET in a moving animal, however, involves technical requirements with inherent trade-offs. Typical FRET measurements require two-color fluorescence imaging at high numerical aperture (NA) (and thus high magnification and limited field of view), which does not allow for capturing body postures and is, therefore, not adequate for tracking freely moving animals. Furthermore, since the fluorescence signal is expected to change in magnitude during motion, registering its exact location along the worm body from frame to frame using fluorescence signal over the entire duration of a free-moving experiment would be highly inaccurate and would not allow us to correlate curvature to FRET signals.

To address these challenges, we needed to decouple the two recordings—fluorescence for FRET and locomotion/body shape for local curvature. We integrated a previously developed tracking system (**Stirman et al., 2011**; **Stirman et al., 2012**) with a separate newly developed FRET imaging capability (**Figure 3**, **Video 1**). This microscope-based platform uses two cameras to enable single-animal tracking with a large enough field of view in bright-field mode (with enough resolution to measure local curvature), while maintaining the animal in the field of view (FOV) in every frame for FRET measurement (with high enough light collecting power and spatial resolution). The bright-field mode uses a near-infrared (NIR) light source to minimize interference with fluorescence imaging (**Figure 3A**, see Materials and methods). NIR and FRET images are precision-aligned at the start of the experiments. The outline of the worm is exactly calculated frame-by-frame from the bright-field image, and the FRET signal can then be mapped to the body (via pre-experiment alignment) and thus accurately correlated with local curvature (see Materials and methods).

The clear contrast between the animal and background in the bright-field imaging mode allows for robust segmentation of the animal posture and quantification of the instantaneous curvature along the body, and thus timings of muscle contractions. To implement fluorescence imaging for FRET, we added to the existing tracking system an optical path that enables two-color imaging and, thereby, measuring FRET signals (**Figure 3A**). The system uses an LCD-projector as a light source, which allows simultaneous and individual control of excitation bands (**Stirman et al., 2011**; **Stirman et al., 2012**). In order to image the donor and acceptor fluorophores in the transgenic animals, we used a specialized filter set in combination with the blue light source from the projector to excite the donor mCFP, and used a beam splitter to simultaneously image the emissions of both the donor mCFP and the acceptor mCitrineFP (see Materials and methods). Fluorescence imaging was then performed with an EMCCD camera, with a frame rate of 20 fps. By using a 5 × objective to give a large FOV to capture the entire worm body, the spatial resolution of measurements is thus limited by the camera, which results in 3.125 µm/px. A key benefit of this system is the low amount of photobleaching due to the illumination at low magnification, allowing for imaging for longer periods of time. Additionally, in our experiments animals expressed a bright GFP marker in the pharynx (using the *myo-2* promoter), which allowed us to maintain the animal in the center of the FOV. Tracking and stage updates were at 10 Hz, which is fast enough to maintain the animal in the FOV at all times, even during fast forward locomotion. By using point set registration, we aligned each fluorescence frame to a bright-field frame, allowing for accurate comparisons of fluorescence signals and body postures (**Figure 3B**). This platform is therefore capable of automatic tracking of worms with simultaneous two-color fluorescence imaging, with a spatial resolution of ~3 µm and temporal resolution of 50 ms. Thereby, this platform enables imaging of freely moving animals expressing FRET moieties and does so over a prolonged period of time (>90 s) to allow for correlational analyses.

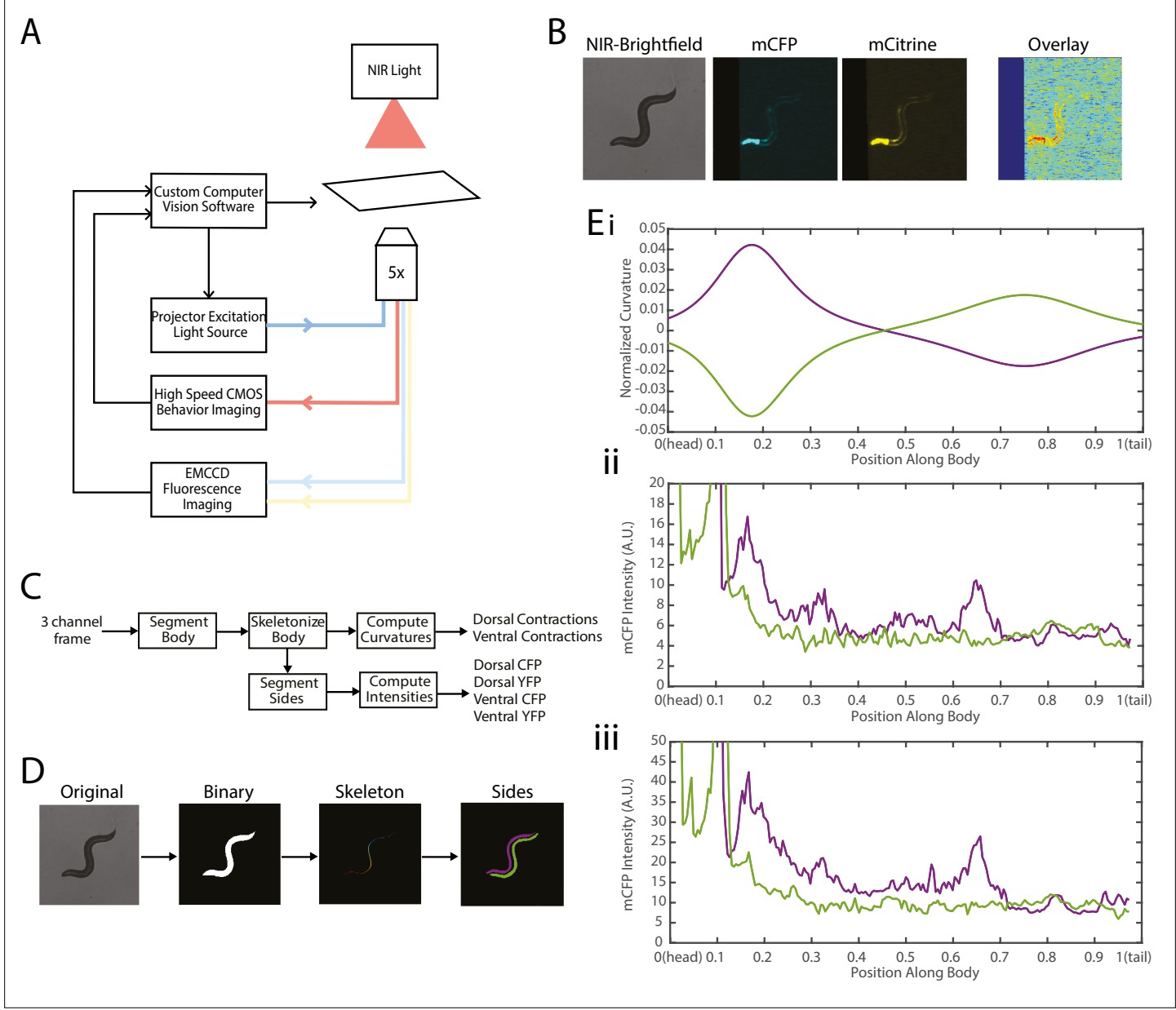

**Figure 3.** Experimental platform for posture tracking and fluorescence imaging. (**A**) Schematic of the platform adapted from *Stirman et al., 2011*, which uses an inverted microscope with a 5 × objective with two optical channels to image in brightfield using NIR and fluorescence imaging using a projector to provide excitation light (see Materials and methods). Tracking is performed by live analysis of a subsample of images, maintaining the pharynx of the animal in the center of the FOV of fluorescence images (see Materials and methods). (**B**) Sample frames acquired using the platform, showing the three channels captured at each time point: NIR brightfield (left), mCFP (center), and mCitrineFP (right). (**C**) The three channels are spatially aligned to allow for accurate extraction and comparisons of muscle contractions and FRET values. (**D**) Schematic of analysis pipeline. The algorithm is performed on each frame, taking as inputs the raw three-channel images. The brightfield image is processed to produce a binary mask of the body outline of the animal. The binary image is then used to create an ordered set of points along the midline of the animal, from head to tail. The midline and body outline are subsequently used to create individual binary mask images of the dorsal and ventral sides of the animal. A tracking algorithm is used to robustly characterize the two sides separately (blue and red). (**E**) Using the midline of the animal, (i) the magnitude of contractions are characterized as computed curvature values, and using the mask images, (ii, iii) the fluorescence intensities are characterized along the length of the animal for both the dorsal and ventral sides (0=head, 1=tail) (green and purple traces represent the two sides). FOV, field of view; NIR, near-infrared.

The online version of this article includes the following figure supplement(s) for figure 3:

**Figure supplement 1.** Tracking of individual sides throughout recordings was performed by comparing angles between three vectors.

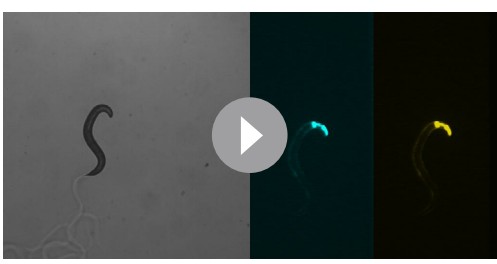

**Video 1.** A freely moving worm being tracked and imaged in dual modality. Two-camera microscopy system enables simultaneous single-animal tracking in bright-field mode (left) and FRET measurement in fluorescent mode (right—mCFP and mCitrine).
https://elifesciences.org/articles/66862/figures#video1

We next developed an automated analysis pipeline to extract FRET signals and muscle contraction states from each frame of the recorded videos, with spatial specificity along the length of the animal accurately and without bias (*Figure 3C*). For each set of frames, we used the bright-field NIR channel image, which has a high contrast between background and foreground, allowing for robust and smooth segmentation of the body outline (*Figure 3D*). The binary image of the body outline was used to create an ordered set of points along the midline of the worm, from head to tail. The midline spline and the body outline were subsequently used to create individual binary mask images of the dorsal and ventral sides of the animal. Ventral and dorsal sides of the worm body were tracked and treated separately and assigned opposite curvature values (*Figure 3Ei*). Muscle contraction states were then characterized by normalized curvature measurements, while FRET signals were computed from extracted mCFP and mCitrineFP emission intensities from the corresponding fluorescence recordings (*Figure 3Eii*, iii, see Materials and methods).

One specific challenge in this analysis was the automatic disambiguation of the two sides (dorsal and ventral) in each frame, which would determine the signs of the curvature. Animals may lie on either the left or right side as they are initially put down on the agar plate, and the animals change locomotion directions drastically throughout recordings. To robustly track the two sides throughout recordings, we computed the angles between the center of mass of each segmented side and the midline of the animal (*Figure 3—figure supplement 1*, and see Materials and methods). This process allowed us to robustly assign consistent labels to the two sides of the animal continuously throughout recordings with an accuracy of >90%, with the majority of incorrectly assigned labels occurring in frames during stage movements.

Changes in fluorescence intensities and curvature over time were visualized in kymographs (*Figure 4A–C*). As expected, most of the time the animals moved forward with a traveling wave propagating backward, and curvature values showing clear and smooth diagonal patterns (*Figure 4A*), similar to recorded kymographs in other studies (*Wen et al., 2012*; *Fouad et al., 2018*; *Gao et al., 2018*). Similarly, mCFP and mCitrineFP emission measurements extracted as a function of both time and distance along the worm body also showed wave-like patterns (*Figure 4B and C*). In comparison to curvature measurements, fluorescence measurements were more variable between animals and along the anterior-posterior axis of the animal, and had a lower signal-to-noise ratio. This is due to the uneven expression level of the transgene in different regions, the diameter of the body varying (and thus introducing optical artifacts differentially along the body), and autofluorescence. Nonetheless, fluorescence measurements seemed to reflect the patterns of muscle contractions. This suggests that FRET and curvature of the muscle are related. Significantly, when examining the curvature and fluorescence signals for each individual segment, we observed oscillatory patterns with similar frequencies for all signals (*Figure 4D and E*). Taken together, the consistency between the fluorescence signals and curvature dynamics of freely moving animals suggests that the relationship between these variables is causal.

## A conformational change occurs in the twitchin kinase region during muscle activity

We next investigated the relationship between the state of muscle contraction and twitchin kinase conformational changes. For this, we quantitatively compared the *C. elegans* strains GB282 and GB284, respectively, overexpressing the test and control twitchin multi-domain fragments (*Figure 2*). The twitchin constructions in this work have been designed to yield maximal FRET signal in the absence of stretch as the protein segments will then be compacted, while the FRET signal will be reduced upon stretch if parts of the twitchin segment undergo force-induced unfolding. Specifically, a positive FRET signal results from donor fluorophores (mCFP) exciting acceptor fluorophores (mCitrineFP), which

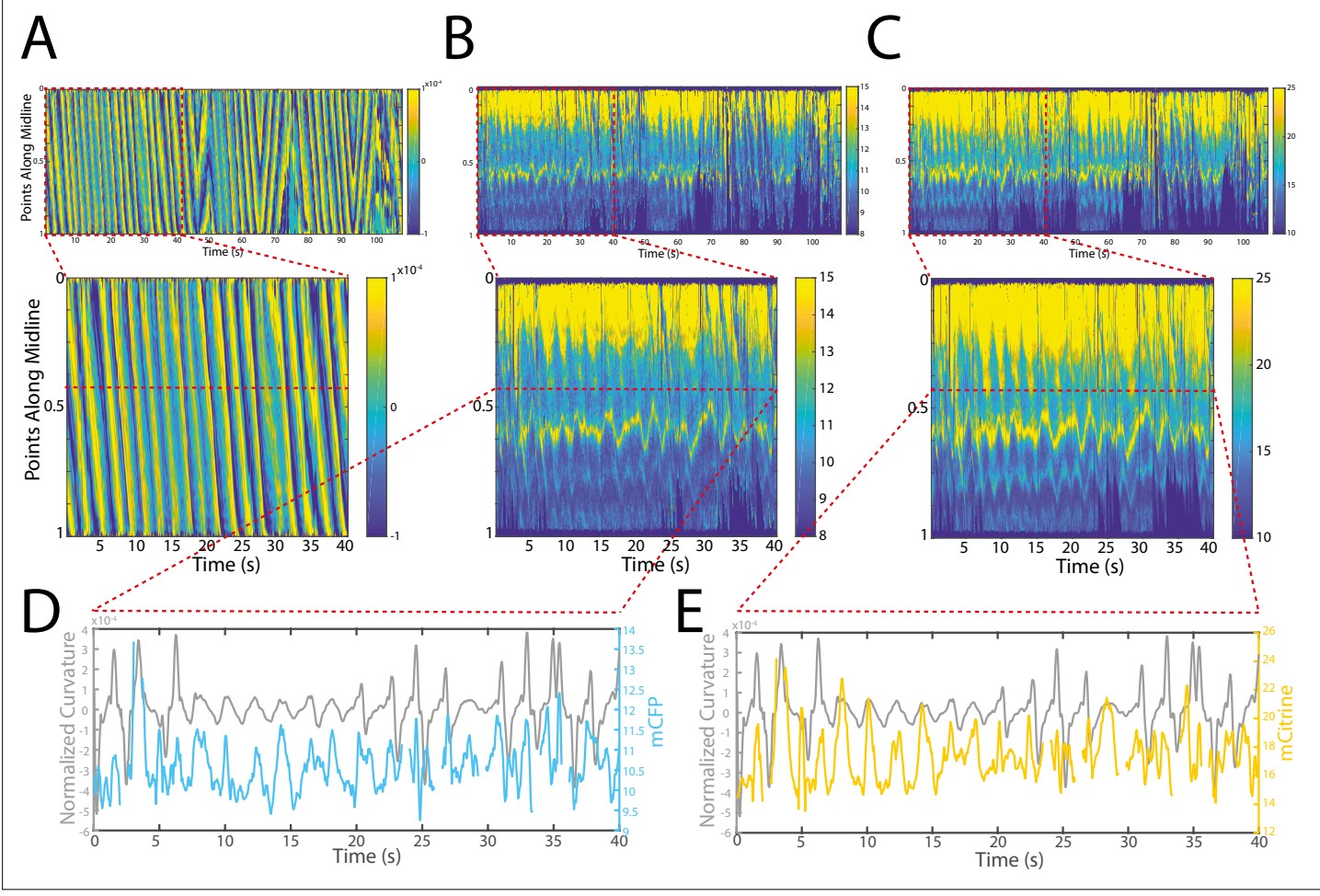

**Figure 4.** Tracking platform and analysis pipeline provide precise spatio-temporal measurements of normalized curvature and fluorescence intensities. (**A**) Kymograph of sample measurements of curvature (C(x,t)) for one side during a recording, where y-axis represents the position along the anterior-posterior axis of the animal, and x-axis represents time. Colormap indicates the computed curvature value. (**B, C**) Kymographs (**B**) of mCFP and mCitrine fluorescence measurements (**C**) in the same side for the recording in (**A**), using the same y-axis and x-axis. Colormap indicates the quantified fluorescence emission. (**D, E**) Sample traces of normalized curvatures (black), and mCFP (blue) in (**D**), and mCitrine (yellow) in (**E**), intensities for a given point along the animal.

typically results in intensity decreases for the donor and increases for the acceptor. In this work, stretch is only expected to affect significantly the conformation of the test construction in strain GB282 and, therefore, to result in larger variations of its FRET signal (ΔFRET) in function of muscle contraction. The inextensible control (GB284) was designed to report on inherent locomotion artifacts, noise, and effects derived from internal sarcomere rearrangements during motion; this ensures that the differences in the measurements between strains can be attributed to conformational changes in twitchin kinase. In addition, changes in signal were obtained in this study by averaging thousands of measurements (see Materials and methods), which would remove non-systematic variability in the observations, such as FRET differences resulting from dynamic orientations of fluorophores.

We first imaged the control strain GB284 (*Figure 2C*), where the FRET fluorophores directly flank the inextensible domain Ig28. To examine the resulting signal, we computed FRET signals normalized by overall changes in intensities (see Materials and methods). This enabled us to compare relative changes in mCFP and mCitrine emissions, while accounting for overall increases in quantified emission values during the movements. In other words, we sought to compare the extent of mCitrineFP fluorescence increase relative to the increase in mCFP fluorescence. It is important to note that because of the undulatory motion that *C. elegans* exhibits when moving on agar, and because the body presses against agar to create a groove during motion, it is experimentally difficult, if not

impossible, to completely eliminate optical and motion artifacts even with the ratiometric nature of the FRET measurement. Although the expectation was that the FRET signal from the control GB284 would remain relatively constant, we observed FRET variations, including an increase in both mCFP and mCitrineFP intensities during muscle contractions (*Figure 5A–D*). We interpreted the variation to result from optical nonlinearities in the recorded emission intensities, instrument artifacts due to hardware motoring during worm tracking and/or possible inter-molecular interactions between FRET moieties in adjacent molecules in the sarcomeric lattice. Indeed, a likely explanation for the observed increase in both mCFP and mCitrineFP intensities during muscle contractions is that fluorophores become more densely packed in space during contractions, leading to increased quantifications of fluorescence emissions as a result of locomotion. Changes in donor and acceptor fluorescence intensity also correlated with the computed normalized FRET signal, where diagonal patterns similar to curvature were also observed. In brief, the control strain reported on all changes in the FRET signal that were due to causes other than the stretch-induced unfolding of the twitchin chain.

Next, we examined the test strain GB282. As in GB284, we also observed increases in both mCFP and mCitrine intensities during muscle contractions (*Figure 5E–G*) and signal normalization revealed a traveling wave (diagonal patterns) similar to curvature (*Figure 5H*), indicating where the mCitrineFP intensity increase dominated over that of mCFP. Compared to GB284, GB282 showed statistically significant larger changes in the overall normalized FRET signal (compare *Figure 5D and H*). Since the relative distances between fluorophores introduced in test and control twitchin segments were similar and the sarcomere context of both constructs was also comparable (*Figure 2*), the larger change in the normalized FRET signal for GB282 with respect to the inextensible control GB284 suggests that FRET fluorophores flanking the kinase domain undergo a change in their mutual distancing during muscle function that is detectable over other sources of signal variation. This result agrees with expectations and points to a stretch-induced conformational change near the kinase domain of twitchin during muscle activity. Moreover, the lower range in FRET signal (ΔFRET) in the control GB284 was as expected and confirmed that the higher level of expression of the fusion protein in GB284 versus the tester strain GB282 (*Figure 5—figure supplement 1*) did not affect the result.

We next quantitatively compared the difference in FRET dynamics in relation to contraction-relaxation cycles between GB282 and GB284. We characterized the cross-correlations between computed FRET signals and curvature measurements, using multiple recordings and treating each point along a worm as an independent sample (*Figure 5—figure supplement 2A, B*). Note that this analysis was not possible without accurate tracking of each point along the AP axis of the worm body. For both strains, the majority of samples have a maximum correlation at a delay of t=0, suggesting that there is no observable delay between FRET signals and curvature as measured in our imaging platform. This observation indicates that the response time of our instrumentation is appropriate and that it did not introduce experimental artifacts. Moreover, the periodic patterns observed in the cross-correlation plots suggest that the observed positive correlation is indeed due to locomotion, and not arbitrary artifacts. As there is no delay between these signals, we computed correlation coefficients between fluorescence signals and curvature values to quantitatively compare relationships in the two strains. As expected, we observed positive correlation coefficients for both mCFP and mCitrineFP intensities, as well as FRET, in both strains (*Figure 5—figure supplement 2C*). Importantly, the correlation coefficients for mCFP are comparable in GB282 and GB284 (*Figure 5—figure supplement 2C*, left), whereas correlation coefficients for mCitrineFP and computed FRET are significantly higher for GB282 as compared to GB284 (*Figure 5—figure supplement 2C*, middle and right). This suggests that putative intermolecular sarcomeric effects or other unaccounted sources could only partially explain the variations in FRET in GB282 during muscle function and that a significant portion of the change in FRET signal in GB282 must arise from conformational changes around the kinase domain.

To further quantify the measured FRET signals during the contraction-relaxation mechanical cycle of the sarcomere, we sought to compare the average measured FRET signals during cycles of muscle contraction. In order to align all our data to muscle activity cycles, we performed an automated analysis that searched our dataset for peaks and valleys of curvature measurements. For this analysis, we defined the muscle contraction using phase angles with 0–360° being a one complete contraction-relaxation cycle, and 0° (and being 360°) 'straight' segments. If one considers body wall muscle cells on the ventral side of a 'straight' worm, 0–90° of curvature represents muscle contracting until it

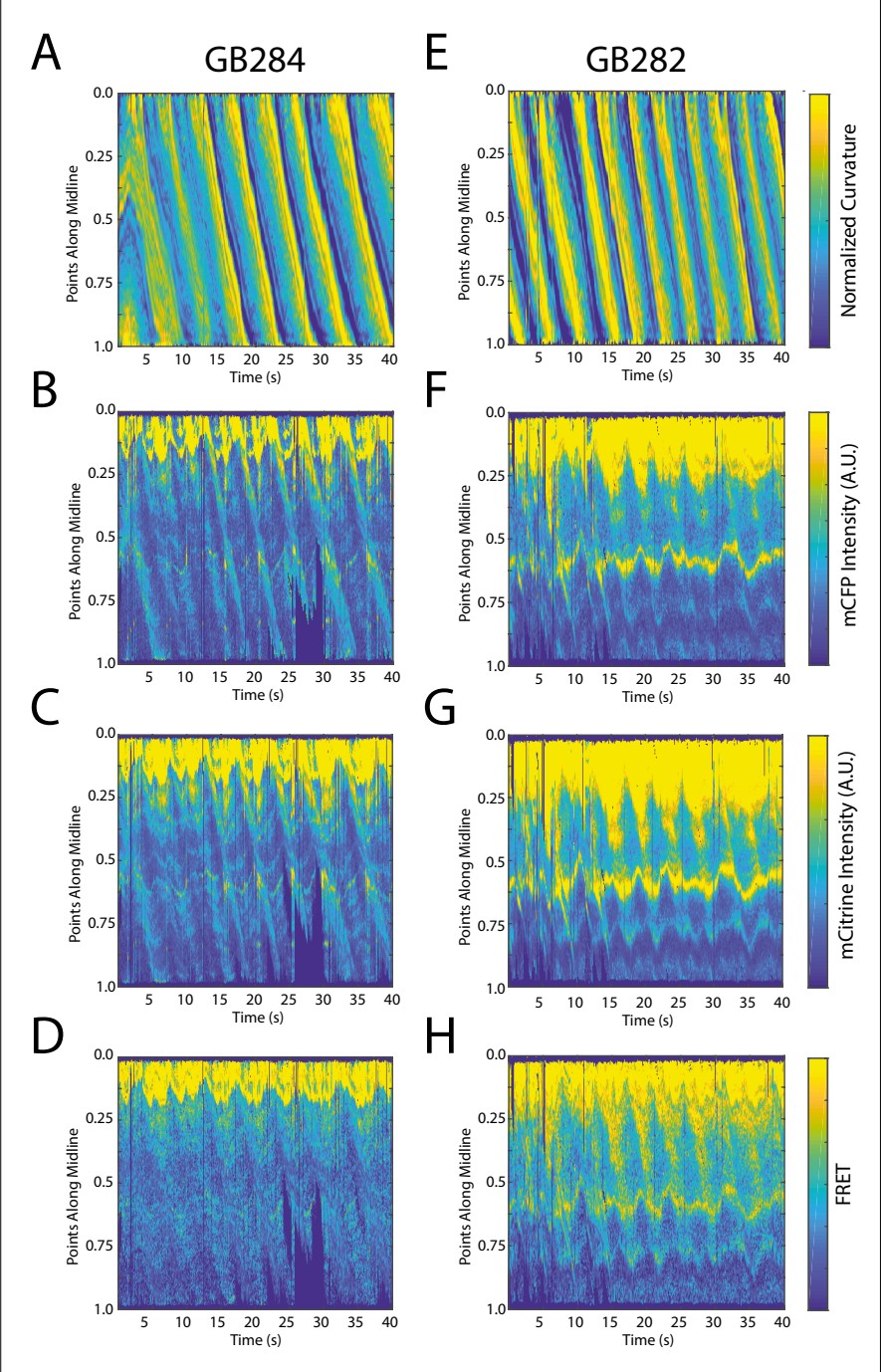

**Figure 5.** FRET signals for GB282 and GB284 are qualitatively correlated with muscle contractions. (A–D) Sample measurements for GB284 of curvature (**A**), mCFP (**B**), mCitrine (**C**), and FRET (**D**) of a recording for a representative worm. (E–H) Sample measurements for GB282 of curvature (**E**), mCFP (**F**), mCitrine (**G**), and FRET (**H**) of a recording for a representative worm. Fluorescence intensities and FRET quantifications are normalized as described in Materials and methods.

The online version of this article includes the following figure supplement(s) for figure 5:

**Figure supplement 1.** Higher level of expression of the fusion protein from the control GB284 as compared with the tester GB282.

**Figure supplement 2.** Statistical analysis reveals correlation between muscle contractions and FRET activity.

maximally contracts and the maximal bent state or positive curvature of the animal is achieved; from

90° to 180°, that muscle is relaxing until the 'straight' state of the worm returns; from 180° to 270°, that muscle continues to relax until maximal negative curvature is achieved; from 270° to 360°, that muscle contracts again to return to the 'straight' worm.

Using this automated analysis, computed curvature values between GB282 and GB284 do not appear significantly different, indicating that the genetically modified sarcomeres in both strains function similarly (*Figure 6A*). We then aligned computed FRET signals to the time points in the mechanical cycle for both strains. Both strains exhibit increases in FRET during muscle contractions and decreases during muscle relaxations, so that the changes in signal are motion dependent in both cases. However, the magnitude of the changes is significantly higher for GB282 (*Figure 6B*).

We compared changes in FRET in four separate sections of the muscle contraction cycle: partially relaxed (partially contracted) to fully contracted (0–90°), fully contracted to partially relaxed (90–180°), partially relaxed to fully relaxed (180–270°), and fully relaxed to partially contracted (partially relaxed) (270–360°). For each part of the muscle contraction cycle, the magnitudes of the changes in computed FRET were significantly higher for GB282 compared to those for GB284 (*Figure 6C*). Additionally, changes in FRET and curvature were positively correlated for all parts of the muscle contraction cycle, with significantly higher correlation coefficients for GB282 compared to GB284 (*Figure 6D*). We performed the same analysis with the quantified emission values for the individual fluorophores, and did not observe consistent increases in change and correlations in all parts of the cycle between GB282 and GB284 for the individual fluorophores (*Figure 2—figure supplement 1*). Most importantly, as is seen in *Figure 6B*, the FRET signal decreases as the muscle relaxes (from 90° to 270°), and this decrease is much stronger for GB282 than for the control GB284. Given this difference in magnitude in the FRET decline between test (GB282) and control (GB284), and that both the test and control constructs had the same backbone and chemical composition and a similar sarcomeric localization, we concluded that a large portion of the change in the FRET signal from GB282 must arise from the increased extension of the FRET moieties, and again not simply from sarcomere lattice effects or experimental artifacts, and this occurs during muscle relaxation.

## Validation of results in a strain with labeled full-length twitchin expressed from its endogenous gene

The observed FRET signal in GB282 arose from multi-domain twitchin segments independently expressed in muscle and subsequently localized to the sarcomere. There was a potential risk that the location of these segments in the sarcomere did not fully mimic the native context of the endogenous kinase region in twitchin. In order to ensure that the changes in FRET in this study report on the physiological response of twitchin kinase in its native sarcomeric context, we created *C. elegans* strain GB287. This strain was developed using CRISPR/Cas9 gene editing to insert coding sequences for the FRET moieties into the endogenous *unc-22* (twitchin) gene, in the same positions as in the transgenic line GB282. The GB287 strain expressed full-length ~750 kDa twitchin with FP proteins flanking the kinase domain and thus, their sarcomeres carried embedded a FRET reporter system. Twitchin expression in this strain is at endogenous levels as the native *unc-22* (twitchin) promoter was utilized. The GB287 worms expressed the tagged twitchin in the appropriate cells (body wall and pharyngeal muscle) (*Figure 7A*), and this tagged twitchin localized properly to sarcomeric A-bands (*Figure 7B*). In addition, GB287 animals displayed normal locomotion. This strain allowed identical experiments to those performed on GB282 and GB284, but with no risk of artifacts resulting from overexpression or fragment mislocalization. Because the FRET moieties are expressed from the single copy native *unc-22* (twitchin) gene, the fluorescence emissions were much dimmer than those from the transgenic lines, leading to increased noise in quantified FRET signals. Nevertheless, when computing cross-correlations between curvature and quantified FRET signals in GB287, we observed clear positive correlations with near zero delay (*Figure 7—figure supplement 1*). Additionally, when comparing FRET changes during muscle contraction cycles, we observed a similar trend to GB282, showing positive correlations between FRET changes and curvature for all phases of the cycle (*Figure 7D–F*). Importantly, the largest and most continuous decrease in FRET occurs during muscle relaxation, in this case from ~140° to 225° (*Figure 7D*), as compared to the large decrease in FRET occurring throughout muscle relaxation (90–270°; *Figure 6B*) seen with the transgenic GB282. Notably, the magnitude of the change in the normalized signal (*Figure 7D*) is higher in GB287 than GB282. This is consistent with the notion that in GB282 twitchin fragments are overexpressed and incorporated

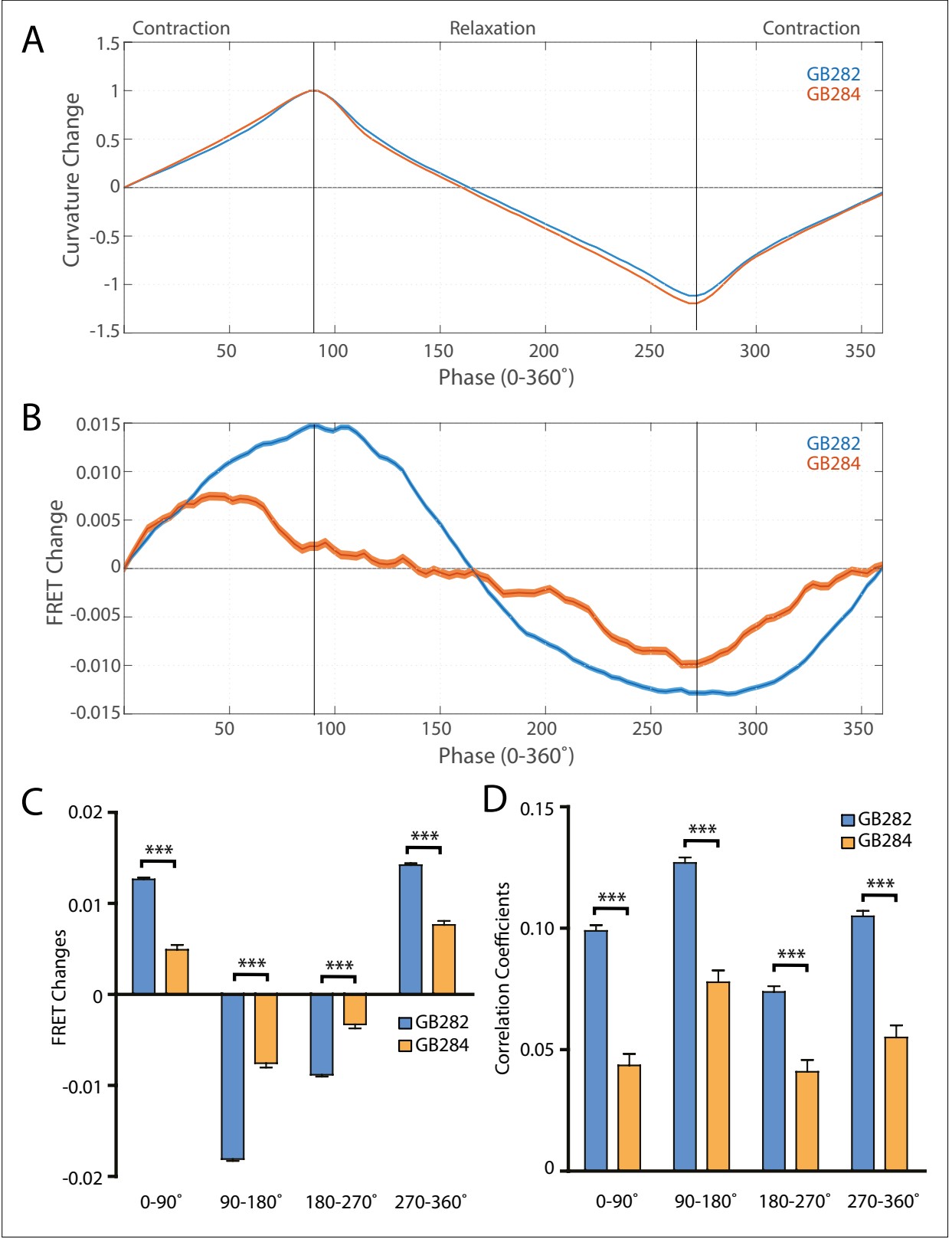

**Figure 6.** Changes in FRET during muscle contraction cycles suggest a conformational change. (**A**) Average normalized curvature values aligned to cycles of muscle contractions, with 90° representing maximum contraction, and 270° representing maximum relaxation, for GB282 (blue, n≥13,840 cycles, 30 animals) and GB284 (orange,n≥11,707 cycles, 30 animals). (**B**) Average FRET value changes for GB282 (blue) and GB284 (orange) aligned to same cycles of muscle contractions as in (**A**). Note that FRET diminishes during muscle relaxation (90–270°). (**C**) Average FRET value changes for GB282

*Figure 6 continued*

and GB284 for each part of a muscle contraction cycle. Statistical comparisons of changes in FRET show statistically higher changes in magnitude for GB282 in comparison to GB284 (Student's t-test, n≥11,707 cycles, 30 animals). ***p≤0.001. (**D**) Average correlation coefficients between FRET value changes and curvature for GB282 and GB284 for each part of a muscle contraction cycle. Statistical comparisons of changes in FRET show statistically higher correlation coefficients for GB282 in comparison to GB284 (Student's t-test, n≥11,707 cycles, 30 animals). ***p≤0.001. All normalized curvature and FRET values were quantified as described in Materials and methods.

in the sarcomere at over-endogenous levels, so that speculatively intermolecular FRET could occur among neighboring molecules, potentially masking the true intramolecular FRET changes. In GB287, where twitchin filaments are at endogenous levels, such intermolecular effects might be decreased, so that the signal variation reports here more directly on intramolecular conformational changes. Altogether, the observed changes in FRET signal in GB287 reproduce the results obtained for GB282. We concluded that the changes must arise from changes in the intramolecular distance in FRET moieties around the kinase domain. Thereby, the signals can be attributed to conformational changes local to the kinase domain of twitchin and enabled by the mechanical liability of the NL and/or CRD tails.

## Discussion

Kinases of the twitchin/titin family are integral components of elastic, sarcomeric filaments that are embedded within the lattice of overlapping thick and thin filaments. These cytoskeletal kinases experience pulling forces during muscle activity, forces which are speculated to induce the conformational unfolding of their inhibitory tail extensions (*von Castelmur et al., 2012*; *Gräter et al., 2005*; *Puchner et al., 2008*; *Puchner and Gaub, 2010*). Mechanically induced changes in these regulatory segments have been speculated to control kinase activity, the recruitment of sarcomeric proteins to kinase-based signalosomes (*von Castelmur et al., 2012*; *Puchner et al., 2008*; *Lange et al., 2005*; *Mayans et al., 2013*) and even to regulate kinase post-translational targetability (*Bogomolovas et al., 2021*). This brings attention to these kinases as likely mechanosensory cytoskeletal nodes that contribute to mediate the response of the myofibril to mechanical stress. However, the occurrence of stretch-unfolding in cytoskeletal components in vivo at physiologically relevant forces remains unproven. Here, we sought to monitor conformational changes in twitchin kinase in working muscles using live, freely moving nematodes. To overcome the technical challenges of this goal, we applied an integrative approach, where a transgenic nematode that expresses twitchin kinase flanked with FRET moieties was generated, and a tracking platform was developed with a robust analysis pipeline for precise measurements of FRET signals in freely moving animals. We observed significant periodic changes in FRET signals during muscle activity in test over control *C. elegans* strains, with a reduction in the FRET signal during muscle relaxation (*Figure 6B*), in agreement with the anticipated conformational extension of TwcK's flanking autoinhibitory tails (*Figure 1D*).

In our methodology, movement effects influenced fluorescence measurements in freely moving animals. First, a measured rise in fluorescence during muscle contraction relative to the slack muscle state appeared to indicate an increased packing density of fluorophores in the contracted sarcomere. Additionally, optical aberrations due to the nearby cuticle or agar likely contributed significantly to our measurements. We used two strategies to overcome movement effects and artifacts in our measurements. First, by using FRET, we computed ratiometric values that were normalized to the sum of the two fluorophores. Although this was effective in removing a large portion of such movement products, it only removed linear effects. Second, we compared test and control strains expressing the same twitchin fragment but containing FRET moieties at different locations. We showed that both fragments incorporated equally in the sarcomere, and thus context-dependent effects—for example, potential FRET arising from intermolecular lattice packing effects in the sarcomere—can be expected to be similar in both strains. As the change in FRET signal is substantially larger in the test strain than in the inextensible control, we reasoned that the increased FRET change in the test strain reflects a change in the position of the FP labels indicative of a conformational extension of the sequences flanking the kinase domain, which exceeds any other movement effects. Thus, by using these two strategies, we were able to confirm with statistical significance that conformational extensibility at the TwcK locus takes place at physiological forces in vivo during muscle activity. It must be noted, however, that since the FRET method applied in this study yields an *average* signal, to which a large number of

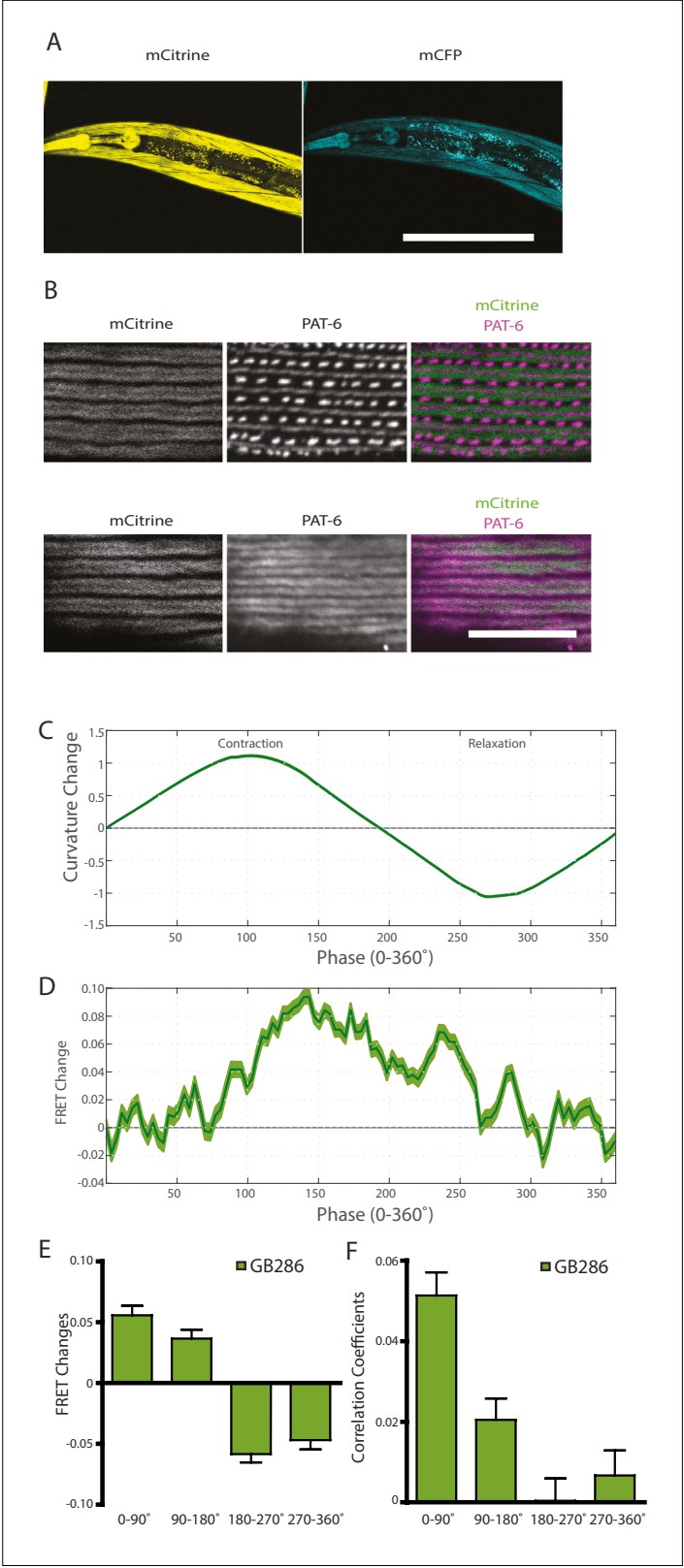

**Figure 7.** Nematodes expressing full-length twitchin with FRET moieties around the kinase domain (GB287) phenocopies the transgenic line (GB282). (**A**) CRISPR/Cas9 gene-edited strain (GB286) expresses endogenous full-length twitchin with FRET moieties in the appropriate cells (body wall and pharyngeal muscle). This twitchin fusion can be imaged either for the mCitrine or mCFP moiety. Scale bar, 100 μm. (**B**) The twitchin FRET fusion protein

*Figure 7 continued on next page*

*Figure 7 continued*

localizes to the proper location in sarcomeres, A-bands. The top row shows co-imaging of the mCitrine signal with immunostaining for PAT-6 (α-parvin), a marker for the base of M-lines and dense bodies (Z-disks). The bottom row show co-localization of the mCitrine signal and immunostaining for twitchin. Scale bar, 10 μm. (**C**) Average normalized curvature values aligned to cycles of muscle contractions, with 90° representing maximum contraction, and 270° representing maximum relaxation, for GB287 (n≥5953 cycles, 30 animals). (**D**) Average FRET value changes for GB287 aligned to same cycles of muscle contractions as in (**C**). (**E**) Average FRET value changes for GB287 for each part of a muscle contraction cycle (n≥5953 cycles, 30 animals). (**F**) Average correlation coefficients between FRET value changes and curvature for GB287 for each part of a muscle contraction cycle (n≥5953 cycles, 30 animals).

The online version of this article includes the following figure supplement(s) for figure 7:

**Figure supplement 1.** Cross-correlation analysis between quantified curvature and FRET values for GB287 (n=6225 midline points, 30 animals).

individual molecules contribute, the signal does not inform on individual changes in individual molecules. Therefore, no information can be inferred on whether unfolding might occur synchronously or asynchronously in the sarcomere.

The methodology that we have developed here could be applied to obtain in vivo evidence for the presumed elasticity of other muscle proteins, especially giant proteins of the twitchin/titin family. For example, the giant protein, UNC-89, homologous to both obscurin and SPEG, has a 647-residue region lying between two protein kinase domains. This region is likely to be a random coil and acts as a highly elastic spring in AFM pulling experiments in vitro (*Qadota et al., 2020*). Although in-frame deletion of this region results in sarcomeric disorganization, whether this region behaves elastically in vivo is not known. Thus, the placement of FRET pairs by CRISPR in or around this region could be used to study its behavior with respect to the contraction-relaxation cycle. Such a strategy could also be used to study the 2.2 MDa nematode protein TTN-1, which contains multiple segments which are speculated to be highly elastic (*Flaherty et al., 2002*; *Forbes et al., 2010*).

As removal of the NL and/or CRD sequences releases the inhibition of TwcK phosphor-transferase activity (*von Castelmur et al., 2012*), our data bring support to the view that sarcomere activity elicits the stretch-induced activation of TwcK. As our data indicate that the kinase activity diminishes during muscle relaxation, it might be deduced that TwcK remains in its inhibited state in inactive muscle. We have reported that a CRISPR-generated twitchin mutant in which TwcK was rendered catalytically inactive resulted in animals that moved faster and, by use of an optogenetic method, their sarcomeres contracted more than those of wild type nematodes (*Matsunaga et al., 2017*). This implied that the phosphotransfer activity of TwcK inhibits muscle activity, most likely phosphorylating a substrate that reduces the strength or duration of muscle contraction. The results reported here lead us to speculate that muscle activity causes the activation of a TwcK signaling pathway. Potentially, once the TwcK substrate has become phosphorylated (i.e., the signal has been launched in the cell), this phosphorylated substrate might stay and accumulate in the myofibril until the signal is abolished by other means (e.g., action of phosphatases, substrate degradation, or recompartmentalization). Such a phosphatase might be part of the sarcomere, perhaps even physically interacting with TwcK. For example, UNC-89 (obscurin) has two protein kinase domains, and this region of the molecule interacts with two classes of protein phosphatases, PP2A (*Qadota et al., 2018*), and SCPL-1 (CTD-type phosphatase) (*Qadota et al., 2008*). Thus, speculatively the TwcK signaling cascade might be active in working muscle. Sarcomere activity (i.e., rate of contraction cycles) might potentially have a titrating effect on the pathway by modulating the cellular levels of phosphorylated substrate. In an inactive muscle, TwcK might remain autoinhibited and, thereby, its signaling pathway shut down. Further insights await determining the physiological substrate for TwcK and the physiological advantage of such reduced muscle contractility. Interestingly, human TK, which shares its multi-domain 3D-architecture with TwcK (*Bogomolovas et al., 2021*), is thought to be a largely inactive pseudokinase (*Bogomolovas et al., 2014*) that serves as scaffold in the recruitment of turn-over protein factors—MuRF E3 ubiquitin ligases and Nbr1/p62 autophagosomal receptors—to the sarcomere (*Lange et al., 2005*). Most recently, a similar stretch-unfolding of TK's NL segment has been proposed to regulate MuRF1-mediated ubiquitination of TK that, in turn, appears to modulate the targetability of this sarcomeric locus by Nbr1/p62 in response to mechanical stress (*Bogomolovas et al., 2021*). Thereby, it is tantalizing to envision

that TK's scaffolding properties and, thus, the targeting of its sarcomeric locus by turn-over factors, could be coupled to muscle inactivity via the elastic properties of TK as part of the onset of the atrophic process. In conclusion, our findings bring support to the view that protein stretch-unfolding of cytoskeletal components is a viable force-sensing mechanism in vivo.

## Materials and methods

### Recombinant protein production

All TwcK sample variants were expressed in *Escherichia coli* Rosetta (DE3) cells (Merck Millipore) and grown in LB media containing 25 µg/ml kanamycin and 34 µg/ml chloramphenicol. Growth was at 37 °C to an $OD_{600}$ of 0.6–0.8 followed by induction of protein expression with 0.5 mM isopropyl β-D-1-thiogalactopyranoside and further growth at 18 °C for approximately 18 hr. Cell pellets were harvested by centrifugation and resuspended in 50 mM Tris-HCl, 500 mM NaCl, 1 mM DTT, pH 7.9 (lysis buffer) supplemented with 20 µg/ml DNase I (Sigma-Aldrich), and one complete EDTA-free protease inhibitor cocktail (Roche) per liter of cell culture. Cell lysis was by sonication on ice, followed by clarification of the lysate by centrifugation at 39,000 ×*g*. Cell lysates were applied to a 5 ml HisTrap HP column (GE Healthcare) equilibrated in lysis buffer containing 20 mM imidazole. Elution of $His_6$-tagged protein was by continuous imidazole gradient, followed by buffer exchange into lysis buffer using PD-10 desalting columns (GE Healthcare), $His_6$-tag cleavage by TEV protease, and subtractive $Ni^{2+}$-NTA purification. The TwcK catalytic domain (Kin) was supplemented with 1 mM $MnCl_2$ and 1 mg Lambda protein phosphatase ($\lambda$ pp) during $His_6$-tag cleavage to remove phosphate groups resulting in inhibitory autophosphorylation (*Williams et al., 2018*). Size exclusion chromatography (SEC) was carried out using a Superdex 200 16/600 column (GE Healthcare) in 50 mM Tris-HCl, 50 mM NaCl, 0.5 mM TCEP, and pH 7.9 (SEC buffer). Purified samples were stored at 4 °C until further use.

### Measurement of phosphotransfer activity by kinase Glo assay

The catalytic activity of the multi-domain Fn-NL-TwcK-CRD-Ig, its FP variants (Fn-mCitrineFP-NL-TwcK-CRD-Ig and Fn-NL-TwcK-CRD-mCyanFP-Ig) and the twitchin kinase domain lacking flanking tails (Kin) was measured by monitoring ATP hydrolysis using the Kinase Glo luminescent assay system (Promega). Assays were performed using 10 µg of protein sample in a 50 µl reaction volume. A peptide derived from chicken myosin light chain (kMLC11–23) with the sequence KKRARAATSNVFS was used as model substrate as previously described (*von Castelmur et al., 2012*; *Williams et al., 2018*; *Heierhorst et al., 1996*). Reactions were carried out in SEC buffer containing 0.5 mM ATP, 1 mM $MgCl_2$, 0.8 mg/ml peptide substrate. and 0.1 mg/ml bovine serum albumin. Samples were incubated at 25 °C for 30 min with gentle shaking before the addition of 50 µl Kinase Glo Max reagent (Promega) and further incubation at 25 °C for 10 min. Control reactions lacking ATP, peptide substrate, ATP/peptide substrate, or kinase protein were included. Luminescence was measured in 96-well Lumitrac plates (Greiner) using a Varioscan Flash Plate Reader (Thermo Fisher Scientific).

### Construction of expression plasmids

Plasmids for expression in *C. elegans* and in bacteria were constructed with PCR using the primers described in *Supplementary file 1*. Details about methods used are described in the *Supplementary file 1*. During cloning, after amplification of fragments by PCR, sequences of fragments were confirmed by DNA sequencing.

### Transgenic worms

To create transgenic worms, a mixture of each twitchin fragment plasmid (10 ng µl⁻¹), together with the transformation marker plasmid, *myo-2p::gfp* (90 ng µl⁻¹), was injected into the gonad of *rde-1(ne219)* (*Tabara et al., 1999*) using standard methods (*Mello and Fire, 1995*). *myo-2p::gfp* expresses GFP exclusively in the pharyngeal muscle. *rde-1(ne219)* is defective in RNAi and prevents the RNAi-induced twitching phenotype of transgenics containing fragments of the *unc-22* (twitchin) gene (*Fire et al., 1991*). Both extrachromosomal arrays were integrated into the genome by UV irradiation (*Mitani, 1995*), with modifications (P. Barrett, personal communication). The resulting strains are as follows: GB282: *rde-1(ne219); sfIs18* [myo-2p::gfp, Ig-Ig-Fn-mCitrine-NL-Kin-CRD-mCFP-Ig-Ig-Ig-Ig-Ig], and GB284: *rde-1(ne219); sfIs19* [myo-2p::gfp, Ig-Ig-Fn-NL-Kin-CRD-Ig-Ig-mCFP-Ig-mCitrine-Ig-Ig].

## Western blot for comparing the level of expression of fusion proteins from GB282 and GB284

The method of *Hannak et al., 2002* was used to prepare total protein lysates from GB282 and GB284. Equal amounts of total protein (diluted 1:20 compared with the original extract) were separated on a 7.5 % SDS-PAGE gel, transferred to nitrocellulose membrane, and stained with Ponceau, and first reacted with antibodies to HA (rabbit monoclonal C29F4 from Cell Signaling Technology, at 1:5000 dilution), and then with anti-myosin heavy chain A (MHC A) (mouse monoclonal 5–6, ascites [a gift from Henry Epstein] [*Miller et al., 1983*], at 1:5000). Goat anti-mouse or goat anti-rabbit IgG conjugated to HRP were used at 1:5000 and 1:10,000 dilutions (from GE Healthcare), respectively, and visualized by ECL. Pairs of GB282 and GB284 extracts were run in quadruplicate on one blot. After scanning in the films, for each lane, the intensity of the HA band was normalized to the intensity of the MHC A band, and the normalized HA band from GB284 was divided by the normalized HA band from GB282 to determine how much more fusion protein was expressed from GB284 versus GB282; from the four pairs, a mean was calculated.

## Creation of a CRISPR strain

A nematode expressing native full-length twitchin with FRET moieties surrounding the twitchin kinase domain was generated commercially (SunyBiotech Corporation) using the following strategy. First, CRISPR/Cas9 was used to produce strain PHX665, *unc-22(syb665)*, in which mCitrineFP was placed N-terminal of NL-Kin, and second, PHX665 was used to place mCFP C-terminal of Kin-CRD, the resulting strain PHX787, *unc-22(syb787syb665)*. As the fluorescence signals from mCitrineFP and, in particular, mCFP, in this strain were quite dim and close in intensity to the autofluorescent gut granules, we crossed PHX787 into *glo-1(zu391)*, resulting in strain GB287, *unc-22(syb787syb665); glo-1(zu391)*. *glo-1* mutants have greatly reduced autofluorescent gut granules (*Hermann et al., 2005*), and indeed GB287 had much less gut fluorescence than PHX787, and thus GB287 was used for all additional experiments.

## Immunolocalization in muscle

To obtain the images shown in *Figure 2B and C*, adult nematodes were fixed as previously described (*Nonet et al., 1993*; *Wilson et al., 2012*). Primary antibodies, anti-HA (mouse monoclonal; H3663; Sigma-Aldrich) and anti-UNC-95 (affinity purified rabbit polyclonal Benian-13; *Qadota et al., 2007*), were used at 1:200 dilution. Secondary antibodies, also used at 1:200 dilution, included anti-rabbit Alexa 488 and anti-mouse Alexa 594 (Invitrogen). Images were captured at room temperature with a Zeiss confocal system (LSM510) equipped with an Axiovert 100 M microscope and an Apochromat 63×/1.4 NA oil objective, in 2.5× zoom mode. To obtain the images of *Figure 7A*, we separately imaged GB287 using the YFP and CFP filter sets available on Olympus confocal model FV1000. To obtain the images of *Figure 7B*, worms were fixed by the constant spring method (*Wilson et al., 2012*), and reacted with either affinity-purified rat anti-PAT-6 (α-parvin) antibodies at 1:100 (*Warner et al., 2013*), or with rabbit anti-twitchin antibodies at 1:100 (*Benian et al., 1993*). Secondary antibodies, used at 1:200 dilution, included anti-rat Alexa 594 and anti-rabbit Cy3 (Invitrogen). Images were captured using the above-described Zeiss confocal system, except that the mCitrine signal was captured using the YFP filter set. The color balances of the images were adjusted by using Adobe Photoshop. We checked the staining of at least three worms for each worm strain.

## Tracking and fluorescence imaging

Tracking and fluorescence imaging were performed on a custom platform adapted from *Stirman et al., 2011*. The platform uses an inverted microscope (Leica-DMIRB) with a low-magnification objective (5 ×) to image freely moving animals. We imaged the animals using NIR light by applying a long-pass filter (715 nm) to the transmitted light path and capture images using a large sensor NIR camera (Basler acA2040-180kmNIR). Posture of the animal was calculated from these images frame-by-frame.

A major improvement was the addition of a separate optical path for imaging the FRET moieties. A FF458-DiO2 Semrock filter cube was used for FRET imaging, which permits the excitation of only the donor fluorophore from the excitation light. The excitation light was delivered using a projector. A DV2 beamsplitter with a 505 nm dichroic mirror and 480/30 m, 535/40 m emission filters were used to simultaneously image mCFP and mCitrineFP. Images were recorded using a Hamamatsu ImageXEM,

with an exposure time of 50 ms. The entire animal was under illumination for FRET imaging. To keep the animal in the FOV in all three channels, we adapted the previous tracking algorithm to track the GFP marker in the pharynx (which is excited by the CFP excitation, albeit less efficiently). Note also that the head region (and therefore the GFP in the pharynx) was excluded during the FRET analysis. The marker was segmented and removed from images. We computed the centroid of the pharynx in terms of x-y pixels on the camera FOV, and based on the position of the computed centroid, a command was sent to a motorized stage to move the animal to the center of the FOV. A desktop computer with an Intel Core i7¬4,790 Processor (8 MB Cache, up to 4.0 GHz) and a 512 GB Solid State Drive and 16 GB RAM was used to process images for tracking. This process allowed for tracking of the animal at an update rate of 10 Hz, while fluorescence images are captured at a frame rate of 19.69 fps, with a resolution of 3.125 µm/px. Bright-field NIR images were captured at a frame rate of 37.61 fps.

## Quantitative behavior and fluorescence analysis

Quantitative values for curvatures and emitted fluorescence intensities were extracted using a custom analysis script as described in *Figure 4*. Because images are captured at a lower frequency in fluorescence mode in comparison to that in bright-field mode, we temporally matched a bright-field image to each fluorescence image. The three images (NIR, mCFP, and mCitrineFP) were spatially aligned by using set point registration of grid images taken prior to the experiment. This resulted in a data set of videos with consistent frame rates in three color channels.

For each time point, we used the bright-field images to compute the body outline and posture. We used this information to quantify muscle contraction states by computing the curvature of each point along the midline of the animal. Each animal was discretized into 100 segments. To precisely quantify fluorescence values as a function of distance along the body, we used the midline of the body outline as a map of distances. We created separate binary mask images for the dorsal and ventral sides of the animal by removing the midline from the body outline. For each point along the midline, a perpendicular line to the tangent of the curve was drawn, and the pixels that overlapped with either the dorsal or ventral sides of the animal were used as a mask to extract fluorescence values. Tracking of individual sides throughout recordings was performed by comparing angles between three vectors: the centroid of each side mask to the closest point in the midline, and the tail to head vector using the midline endpoints (*Figure 3—figure supplement 1*). We used an empirically defined threshold of angle comparison to distinguish the two sides, and discarded frames where the angle differences did not pass a threshold.

## Statistics

### Muscle contraction and fluorescence values

Muscle contraction values were computed as curvature values. Curvature values were computed by first fitting two separate quaternary degree polynomials for the x and y coordinates for the midline of the animal in an image, then computing curvature values along the midline by using the following formula:

$$K = \frac{\left| x' y'' - y' x'' \right|}{(x'^2 + y'^2)^{\frac{3}{2}}}$$

We assigned values of equal magnitude and opposite signs to the dorsal and ventral sides, assigning positive values to contracted states and negative values to stretched states.

FRET values for each point in time and along the body of the animal were computed as follows:

$$FRET(x, t) = (E_{535}(x, t) - E_{480}(x, t))/(E_{535}(x, t) + E_{480}(x, t))$$

This quantification allows for normalization of overall changes in fluorescence. When comparing statistics for individual colors, values are normalized by changes from the minimum quantified value ($\Delta F/F_{min}$).

## Correlation values and cross-correlations
Correlation coefficients between muscle contractions and FRET values were computed using MATLAB. Cross-correlation was computed using normalization to allow the comparison of results between two strains. Experiments were performed on 10 s of animals per genotype and measurements were taken along the body and discretized to ~100 segments per worm. Each segment was tracked with curvatures and FRET measured, and was treated as an independent sample in the correlation analysis. Samples with low values of fluorescence emission were omitted from analysis. Statistical significance was analyzed using Student's t-test.

## Comparisons per muscle contraction cycle
Each cycle of muscle contraction was extracted by searching for local maxima and minima in normalized curvature data. Thresholds were used for peak prominences and widths to minimize false detections of muscle contractions. Time points for each muscle contraction cycle were recorded and used to extract FRET changes as a function of muscle contraction. Changes in FRET for each part of the cycle were computed by subtracting the FRET values at the last index of the cycle from the values at the first index of the cycle. Statistical significance was evaluated using Student's t-test.

## Acknowledgements
The authors gratefully acknowledge support from a Human Frontier Science Program Grant (RGP0044/2012).

# Additional information

### Funding

| Funder | Grant reference number | Author |
| --- | --- | --- |
| Human Frontier Science Program | RGP0044/2012 | Olga Mayans<br>Guy M Benian<br>Hang Lu |

The funders had no role in study design, data collection and interpretation, or the decision to submit the work for publication.

### Author contributions
Daniel Porto, Conceptualization, Data curation, Formal analysis, Investigation, Methodology, Visualization, Writing - original draft; Yohei Matsunaga, Barbara Franke, Rhys M Williams, Hiroshi Qadota, Investigation; Olga Mayans, Conceptualization, Funding acquisition, Investigation, Methodology, Project administration, Resources, Supervision, Writing - original draft, Writing - review and editing; Guy M Benian, Conceptualization, Data curation, Formal analysis, Investigation, Methodology, Project administration, Resources, Supervision, Visualization, Writing - original draft, Writing - review and editing; Hang Lu, Conceptualization, Formal analysis, Funding acquisition, Investigation, Project administration, Resources, Supervision, Visualization, Writing - original draft, Writing - review and editing

### Author ORCIDs
Daniel Porto (ID) http://orcid.org/0000-0002-1021-2467
Rhys M Williams (ID) http://orcid.org/0000-0002-1982-2632
Olga Mayans (ID) http://orcid.org/0000-0001-6876-8532
Guy M Benian (ID) http://orcid.org/0000-0002-8236-3176
Hang Lu (ID) http://orcid.org/0000-0002-6881-660X

### Decision letter and Author response
Decision letter https://doi.org/10.7554/eLife.66862.sa1
Author response https://doi.org/10.7554/eLife.66862.sa2

## Additional files

### Supplementary files

- Supplementary file 1. DNA sequences of the primers used in this work.

- Transparent reporting form

### Data availability

All data generated or analyzed during this study has been deposited to Dryad.

The following dataset was generated:

| Author(s) | Year | Dataset title | Dataset URL | Database and Identifier |
|---|---|---|---|---|
| Porto D | 2021 | FRET imaging and curvature data of freely moving *C. elegans* | https://doi.org/10.5061/dryad.6wwpzgn09 | Dryad Digital Repository, 10.5061/dryad.6wwpzgn09 |

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

## Appendix 1

## Additional methods

### *C. elegans* expression plasmids

Note: the DNA sequences of the primers are described in *Supplementary file 1*.

pPD95.86-HA-UNC-22 Fn-kin-Ig (myo-3p-HA-UNC-22 Fn-kin-Ig): cDNA was amplified by using Twik-10 and TwiK-11 primers and pET28-Fn-kin-Ig (*Greene et al., 2008*) as a template and cloned into pKS-HA8(NheI), resulting in pKS-HA- UNC-22 Fn-kin-Ig. A NheI fragment of pKS-HA-UNC-22 Fn-kin-Ig was cloned into pPD95.86.

pPD95.86-HA-UNC-22 Ig-Ig-Fn-kin-Ig (myo-3p-HA-UNC-22 Ig-Ig-Fn-kin-Ig): cDNA was amplified by using Twik-13 and TwiK-15 primers and RB2 cDNA library and cloned into pKS-HA8(NheI), resulting in pKS-HA-UNC-22 Ig-Ig-Fn-kin-Ig. A NheI fragment of pKS-HA-UNC-22 Ig-Ig-Fn-kin-Ig was cloned into pPD95.86.

pPD95.86-HA-UNC-22 Fn-kin-Ig-Ig-Ig (myo-3p-HA-UNC-22 Fn-kin-Ig-Ig-Ig): cDNA was amplified by using Twik-17 and TwiK-18 primers and RB2 cDNA library and cloned into pKS-HA8(NheI), resulting in pKS-HA-UNC-22 Fn-kin-Ig-Ig-Ig. A NheI fragment of pKS-HA-UNC-22 Fn-kin-Ig-Ig-Ig was cloned into pPD95.86.

pPD95.86-HA-UNC-22 Ig-Ig-Fn-kin-Ig-Ig-Ig-Ig (myo-3p-HA-UNC-22 Ig-Ig-Fn-kin-Ig-Ig-Ig-Ig-Ig): cDNA was amplified by using Twik-19 and TwiK-20 primers and RB2 cDNA and cloned into pGAD-C1, resulting in pGAD-UNC-22 kin-Ig-Ig-Ig-Ig-Ig. NcoI, SalI fragment of pGAD-UNC-22 kin-Ig-Ig-Ig-Ig-Ig was cloned into NcoI, SalI sites of pKS-HA-UNC-22 Ig-Ig-Fn-kin-Ig, resulting in pKS-HA-UNC-22 Ig-Ig-Fn-kin-Ig-Ig-Ig-Ig-Ig. A NheI fragment of pKS-HA-UNC-22 Ig-Ig-Fn-kin-Ig-Ig-Ig-Ig-Ig was cloned into pEGFP-N1 for efficient cloning. A NheI fragment of pEGFP-N1-HA-UNC-22 Ig-Ig-Fn-kin-Ig-Ig-Ig-Ig-Ig was cloned into pPD95.86.

pPD95.86-HA-UNC-22 Ig-Ig-Fn-mCit-kin-mCFP-Ig-Ig-Ig-Ig-Ig (myo-3p-HA-UNC-22 Ig-Ig-Fn-mCit-kin-mCFP-Ig-Ig-Ig-Ig-Ig): cDNA fragment containing mCitrine was created by three PCRs. First PCR: fragments that amplified with primer sets, TwFRET-1/TwFRET-3 or TwFRET-4/TwFRET-2, and pKS-HA-UNC-22 Ig-Ig-Fn-kin-Ig-Ig-Ig-Ig-Ig as a template. Second PCR: Fragment was amplified with primer set TwFRET-1/TwFRET-5 and mixture of pETM-mCitrine and the first PCR product of primer set TwFRET-1/TwFRET-3 as templates. Third PCR: Fragment was amplified with primer set TwFRET-1/TwFRET-2 and mixture of second PCR product and first PCR product of primer set TwFRET-4/TwFRET-2 as templates. The third PCR product was cloned into pBluescript, resulting in pBS-TwFRET-mCitrine. BstEII, EcoRV fragment of pEGFP-N1-HA-UNC-22 Ig-Ig-Fn-kin-Ig-Ig-Ig-Ig-Ig was exchanged with the BstEII, EcoRV fragment of pBS-TwFRET-mCitrine, resulting in pEGFP-N1-HA-UNC-22 Ig-Ig-Fn-mCit-kin-Ig-Ig-Ig-Ig-Ig. cDNA fragment containing mCFP was created by five steps of PCR. First PCR: Fragments that amplified with primer sets, TwFRET-10/TwFRET-11 or TwFRET-12/TwFRET-9, and pKS-HA-UNC-22 Ig-Ig-Fn-kin-Ig-Ig-Ig-Ig-Ig as a template. Second PCR: Fragment was amplified with primer set TwFRET-10/TwFRET-13 and mixture of pETM-mCFP and first PCR product of primer set TwFRET-10/TwFRET-11 as templates. Third PCR: Fragment was amplified with primer set TwFRET-10/TwFRET-9 and mixture of second PCR product and first PCR product of primer set TwFRET-12/TwFRET-9 as templates. Fourth PCR: fragments that amplified with primer sets, TwFRET-14/TwFRET-15 and pEGFP-N1-HA-UNC-22 Ig-Ig-Fn-mCit-kin-Ig-Ig-Ig-Ig-Ig as a template. Fifth PCR: Fragment was amplified with primer set TwFRET-10/TwFRET-15 and mixture of third PCR product and fourth PCR products as templates. The fifth PCR product was cloned into pBluescript, resulting in pBS-TwFRET-mCFP. ScaI, BstXI fragment of pEGFP-N1-HA-UNC-22 Ig-Ig-Fn-mCit-kin-Ig-Ig-Ig-Ig-Ig was exchanged to ScaI, BstXI fragment of pBS-TwFRET-mCFP, resulting in pEGFP-N1-HA-UNC-22 Ig-Ig-Fn-mCit-kin-mCFP-Ig-Ig-Ig-Ig-Ig. A NheI fragment of pEGFP-N1-HA-UNC-22 Ig-Ig-Fn-mCit-kin-mCFP-Ig-Ig-Ig-Ig-Ig was cloned into pPD95.86.

pPD95.86-HA-UNC-22 Ig-Ig-Fn-kin-Ig-Ig-mCFP-Ig-mCit-Ig-Ig (myo-3p-HA-UNC-22 Ig-Ig-Fn-kin-Ig-Ig-mCFP-Ig-mCit-Ig-Ig): For creating Ig-mCFP-Ig fragment, three steps of PCR were used. First PCR: Fragments that amplified with primer sets, TwFRET-21/TwFRET-22 or TwFRET-23/TwFRET-24, and pKS-HA-UNC-22 Ig-Ig-Fn-kin-Ig-Ig-Ig-Ig-Ig as a template. Second PCR: Fragment was amplified with primer set TwFRET-21/TwFRET-13 and mixture of pETM-mCFP and first PCR product of primer set TwFRET-21/TwFRET-22 as templates. Third PCR: Fragment was amplified with primer set TwFRET-21/TwFRET-24 and mixture of second PCR product and first PCR product of

primer set TwFRET-23/TwFRET-24 as templates. The third PCR product was cloned into pBluescript, resulting in pBS-Ig-mCFP-Ig. For creating Ig-mCitrine-Ig-Ig fragment, two-step PCR was used. First PCR: Fragments that amplified with primers TwFRET-25/TwFRET-5 and pETM11-mCitrine as a template or primers TwFRET-26/TwiK-20 and pKS-HA-UNC-22 Ig-Ig-Fn-kin-Ig-Ig-Ig-Ig-Ig as a template. Second PCR: Fragment was amplified with primer set TwFRET-27/TwiK-20 and mixture of two first PCR products as templates. The second PCR product was cloned into pBluescript, resulting in pBS-Ig-mCitrine-Ig-Ig. BamHI, EcoRI fragment of pGAD-UNC-22 kin-Ig-Ig-Ig-Ig-Ig into BamHI, EcoRI site of pBS-Ig-mCFP-Ig, resulting in pBS-UNC-22 Ig-Ig-mCFP-Ig. PstI, SalI fragment of pBS-Ig-mCitrine-Ig-Ig was cloned into PstI, SalI site of pBS-UNC-22 Ig-Ig-mCFP-Ig, resulting in pBS-UNC-22 Ig-Ig Ig-mCFP-Ig-mCitrine-Ig-Ig. NcoI, SalI fragment of pBS-UNC-22 Ig-Ig Ig-mCFP-Ig-mCitrine-Ig-Ig was cloned into NcoI, SalI site of pKS-HA-UNC-22 Ig-Ig-Fn-kin-Ig, resulting in pKS-HA-UNC-22 Ig-Ig-Fn-kin-Ig-Ig-mCFP-Ig-mCit-Ig-Ig. A NheI fragment of pKS-HA-UNC-22 Ig-Ig-Fn-kin-Ig-Ig-mCFP-Ig-mCit-Ig-Ig was cloned into pEGFP-N1 for efficient cloning. A NheI fragment of pEGFP-N1-HA-UNC-22 Ig-Ig-Fn-kin-Ig-Ig-mCFP-Ig-mCit-Ig-Ig was cloned into pPD95.86.

## Bacterial expression plasmids

pETM11-UNC-22 kin: This plasmid was described previously (*von Castelmur et al., 2012*).

pET28-UNC-22 Fn-kin-Ig: This plasmid was described previously (*Greene et al., 2008*).

pETM11-UNC-22 Fn-mCit-kin-Ig: cDNA fragment was amplified by using primers TwFRET-6/TwFRET-7 or TwFRET-8/TwFRET-9 and pEGFP-N1-HA-UNC-22 Ig-Ig-Fn-mCit-kin-Ig-Ig-Ig-Ig-Ig as a template. Amplified fragment was cloned into pBluescript, resulting in pBS-TwFRET-67 and pBS-TwFRET-89. BamHI, EcoRV fragment of pBS-TwFRET-67 was cloned into BamHI, EcoRV sites of pBS-TwFRET-89, resulting in pBS-TwFRET-6789 (Fn-mCit-kin-Ig). BsmBI, KpnI fragment of pBS-TwFRET-6789 was cloned into pETM11 NcoI, KpnI sites, resulting in pETM-UNC-22 Fn-mCit-kin-Ig.

pETM11-UNC-22 Fn-kin-mCFP-Ig: cDNA fragment was amplified by using primers TwFRET-6/TwFRET-7 and pKS-HA-UNC-22 Ig-Ig-Fn-kin-Ig-Ig-Ig-Ig-Ig as a template or TwFRET-8/TwFRET-9 and pKS-HA-UNC-22 Ig-Ig-Fn-kin-Ig-Ig-mCFP-Ig-mCit-Ig-Ig as a template. Amplified fragment was cloned into pBluescript, resulting in pBS-TwFRET-67WT and pBS-TwFRET-89mCFP. BamHI, EcoRV fragment of pBS-TwFRET-67WT was cloned into BamHI, EcoRV sites of pBS-TwFRET-89mCFP, resulting in pBS-TwFRET-6789mCFP (Fn-kin-mCFP-Ig). BsmBI, KpnI fragment of pBS-TwFRET-6789mCFP was cloned into pETM11 NcoI, KpnI sites, resulting in pETM-UNC-22 Fn-kin-mCFP-Ig.

