## [Decision Letter]

**Acceptance summary:**

This paper will be of broad interest, particularly to mechanobiologists and muscle scientists. The authors investigated how muscle contraction might be linked to the mechanical activation of a kinase domain in a large structural protein in a living animal. They developed new imaging methods to relate molecular-level mechanical events to the motion of the whole organism, *C. elegans*. The work points to a potentially new regulatory mechanism via a mechanically sensitive kinase. The study combines imaging of the moving live animal with FRET measurements to show that twitchin kinase activation is coupled to muscle contraction. The necessary development of an imaging and image analysis platform that combines information on locomotion with fluorescence output is an important advance in itself. The conclusions might represent ground-breaking work into cytoskeletal signaling mechanisms.

**Decision letter after peer review:**

Thank you for submitting your article "Conformational changes in twitchin kinase in vivo revealed by FRET imaging of freely moving *C. elegans*" for consideration by *eLife*. Your article has been reviewed by 3 peer reviewers, and the evaluation has been overseen by a Reviewing Editor and Olga Boudker as the Senior Editor. The following individuals involved in review of your submission have agreed to reveal their identity: James R Sellers (Reviewer #1); Isaac Li (Reviewer #2); Mathias Gautel (Reviewer #3).

Essential revisions:

All reviewers agreed that this is a high-quality significant paper that requires primarily editorial revisions. Below I have compiled questions, comments, and suggestions that the reviewers have made.

(1) The authors address the importance of discriminating between the desired intramolecular FRET signal versus any intermolecular FRET, which, given the tight molecular packing of the myofilament lattice, is an important concern to be addressed. Their approach was to introduce the FRET fluorophores into two different positions along the transgenic reporter construct and ultimately, for the test construct, in the gene edited nematodes. However, for force to act across the kinase region, it would have to be bound to two components of the myofilament lattice moving with respect to each other (explaining sufficient conformational changes to account for the FRET changes and induce an open kinase conformation) by its N- versus C-terminal regions. Therefore, a C-terminally integrated FRET may be in a different molecular and mechanical environment than the regions immediately around the kinase domain. Ideally, to assess the level of intermolecular FRET, two constructs with Donor or Acceptor fluorophore only, but in their correct positions for the complete sensor, are co-expressed. Any FRET signal detected and any changes in FRET would then be attributable to intermolecular FRET occurring at the site of the sensor due to spatial proximity of the donor and acceptor fluorophores.

(2) Figure 5 B and C indicate that expression of the transgenic FRET sensor seems higher in GB282 than in GB284, and this appears to correlate with higher FRET signals. Has the FRET signal been corrected for expression levels or donor-acceptor intensity?

(3) Given the major achievement of generating the GB287 (or GB286?) genome-edited nematode line, it is surprising that its characterisation is terse and the results are banished to the supplements. Could FRET videos of this line be shown? Where in the sarcomere does the FRET probe localise?

(4) It seems reasonable to postulate that the relationship "between the fluorescence signal and the curvature dynamics … is causal" – but is it due to active force? Would the FRET signal change persist if the worms were paralysed or anaesthetised, e.g. with BDM or a sodium channel blocker, e.g. tricaine? In other words, is this a geometric (lattice packing) or biomechanical (active forces) effect, as far as this can be separated?

(5) It is intriguing that the largest decrease in FRET occurs in relaxed sarcomeres, implying that the kinase region would be stretched preferentially when the myosin heads are relaxed. Again, could this hypothesis be tested by interfering with the actomyosin cycle, for example with para-nitroblebbistatin? Could the authors propose, with data from point 3, a hypothetical arrangement of twitchin in the myofilament lattice that would satisfy these observations?

(6) The propagating patterns in the kymographs of mCFP and mCit (Figure 5BCFG) are puzzling. The authors contributed it to inherent locomotion artifacts, noise and internal sarcomere rearrangement during motion. While some of these may be true, could these be image processing artifacts? The authors stated in the method section that the fluorescence intensity at a particular body segment is obtained by drawing a perpendicular line to the midline. The pixels that it intersects will provide the fluorescence intensity. This approach does not seem to account for the fluorophore density change due to tissue compaction and expansion, resulting in overcounting intensity in the inner circle and undercounting at the outer circle – similar to the observed intensity patterns Figure 5BCFG. Can the intensities be normalized by the arclength at the different radii from the center of the curvature?

(7) As related to the previous comment, but more generally, image analysis is a critical and sensitive step towards the interpretation of the fluorescence results. The authors should elaborate if and how errors in the image processing might contribute to the emergence of correlation between FRET and curvature. For instance, the CFP and mCit expression levels vary significantly along the body of the worm (Figure 5) and should be time-invariant. If an error in image processing picks up nearby spatial variations as the worm moves, the detected fluorescence will become time-variant and correlate with the worm's motion. Could this happen with the current algorithm? This is a crucial assessment as it is crucial to ensure the observed small FRET changes (+/- 0.015) are due to molecular stretching and not artifacts of image processing.

(8) The shape and meaning of FRET change in the contraction-relaxation cycles (Figure 7) require further interpretation. The data shows that the extrema and phase of the FRET signal correlate to curvature, and thereby, sarcomere stretching. Is it valid to assume the stretching or relaxing of sarcomeres apply tension directly over each twitchin? Is the binding-unbinding transition of NL to TwcK two-state? If so, would this lead to two-state behaviour in the observed FRET? What can the authors comment on the shape of the FRET-curvature response curve?

(9) The reason behind the small observed FRET change (+/-0.015) requires further clarification. Is it because (1) all FRET sensors changed slightly, or (2) a small fraction of FRET sensors changed from high to low FRET. What is the expected FRET change as depicted in Figure 1D?

(10) The manuscript provides strong evidence of FRET correlating to curvature during the muscle contraction cycle. However, the causality is less clear. Does the contraction force cause the FRET change, or can curvature without any active contraction cause FRET change? For instance, if the worm is dead or myosin activity inhibited, will the bending of the worm cause FRET change?

(11) FRET was used as a proxy for kinase function in some of the discussions. Although this may be the case as expected based on crystal structures, it is not demonstrated, i.e. when the mCit-TwcK-mCFP is stretched, it is unknown whether the kinase regains activity. The current controls for TwcK activity (Figure 1C) only demonstrated the lack of kinase activity for TwcK-FP but not mCit-TwcK-mCFP. There could also be many reasons for the lack of activities, e.g.: (1) NL peptide autoinhibition, (2) misfolding of TwcK domain when tagged by fluorescent proteins, (3) steric-hindrance of target peptide binding by the fluorescent proteins. Can the kinase activity in the presence of force be experimentally demonstrated? I recognize that this may be a very challenging experiment, and outside of the scope of the current manuscript, it should be discussed. In any case, FRET is still valid as a proxy for tension through TwcK. However, arguments based on kinase function during FRET change can only be inferred indirectly. These points should be made more explicitly in the article.

(12) Is there any existing estimation on the magnitude of force required to peel the NL domain off from TwcK? I think some estimates of the expected dynamic range of FRET in response to force (order of magnitude estimation) could help interpret the result.

(13) Direct comparison of FRET between mCit-TwcK-mCFP (GB282) and mCFP-Ig-mCit (GB284) requires clarification. First, GB282 seems to be expressing the proteins at a higher level comparing to GB284 (Figure 5). Is this true for all worms? Second, it seems that GB284 generally has a lower FRET comparing to GB282. This is puzzling as the authors assumed the Ig domains do not unfold, and hence GB284 should remain at high FRET compared to GB282. Isn't the unfolding of Ig domains a physiological process in muscle function? What is the evidence that Ig domains in the control construct would not unfold? Would it be a better control to remove proteins between the two FRET FPs if this is uncertain? Is maintaining the exact distance between the two FPs crucial?

(14) What causes the expression level difference along the length of the worm? It seems that each part of the worm can contract similarly (Figure 4A), which indicates the presence of muscles and sarcomeres along the entire body.

(15) The authors have made several arguments regarding intermolecular FRET. The Foster distance for fluorescent proteins is only 4-6 nm. Considering that they sit at a particular location on the long polydomain protein, it would require perfect alignment with nearby twitchins at a very high spatial density for intermolecular FRET to occur, which seems unlikely. Do the authors have evidence that intermolecular FRET is indeed happening?

(16) I think it would be beneficial if the authors can show that dorsal and ventral FRET anticorrelate as a validation of their method and strengthen the paper.

(17) Many of the readers will not be familiar with *C. elegans* motility and the manuscript would benefit from having a supplemental video of a moving worm, perhaps captured by each of the imaging modalities.

(18) Similarly, there should be some description of what is the period of the undulating waves.

(19) It took me a while to wrap my head around the kymographs in Figure 4 and subsequent figures and it would be helpful to have a bit more discussion of how they are generated and interpreted. I assume that each of the diagonals in the kymographs represent a single contractive event. Can you average 10 or more of these in cases where the direction and period are very similar to get cleaner signals?

(20) Is there any way to instantaneously paralyze the worms and freeze them in their undulating shapes? If so, could this simplify the imaging?

(21) I think that the data shown on the knock-out/knock-in worms is important and I suggest moving it to the main text. Figure 6 could go to the supplement.

(22) Figure 1D seems only to suggest that the NL domain is only stretched instead of unbinding then stretched from the kinase. Figure 1A is a clockwise 90-degree turn, which was not clear from the illustration. The ATP pocket should be marked in both figures.

(23) Figure 2D: The assessment of distance of 5.3 nm is the distance between the N/C termini of the fluorescent proteins, but not the fluorophores. The two fluorophores' distance and orientation would affect the coupling and FRET between them. Can these values be estimated? In addition, are the observed FRET in vivo similar to FRET observed in vitro using recombinant proteins?

(24) Figure 5: The fluorescence intensity is saturated. Is this a display issue, or is the fluorescence data collected saturated? If display issue, I would recommend a non-linear colour legend to display the full dynamic range. If the collection is saturated, then wherever saturation occurs should be excluded for all analyses.

(25) I do not see where Figure 3E is mentioned in the text.

(26) Figure 4D,E. Describe the colors used for the two traces in each panel.

(27) The gene-edited twitchin line is referred to as GB287 in figure legends and text (e.g. page 15), but as GB286 in supplementary figure S4.

(28) Please correct the typo in Figure 3A: EMCCD Fluorescence Imaging

(29) Please clarify what is meant with "computed" when referring to e.g. "computed change in FRET signal" as the reader might assume the data are simulations not experimental.

*Reviewer #1* (*Recommendations for the authors*):

Many of the readers will not be familiar with *C. elegans* motility and the manuscript would benefit from having a supplemental video of a moving worm, perhaps captured by each of the imaging modalities.

Similarly, there should be some description of what is the period of the undulating waves.

It took me a while to wrap my head around the kymographs in Figure 4 and subsequent figures and it would be helpful to have a bit more discussion of how they are generated and interpreted. I assume that each of the diagonals in the kymographs represent a single contractive event. Can you average 10 or more of these in cases where the direction and period are very similar to get cleaner signals?

Is there any way to instantaneously paralyze the worms and freeze them in their undulating shapes? If so, could this simplify the imaging?

I think that the data shown on the knock-out/knock-in worms is important and I suggest moving it to the main text. Figure 6 could go to the supplement.

I do not see where Figure 3E is mentioned in the text.

Figure 4D,E. Describe the colors used for the two traces in each panel.

*Reviewer #2* (*Recommendations for the authors*):

I enjoyed reading this manuscript and appreciated the transparent discussion of many potential issues. In addition to the issues pointed out in the public review, addressing the following points could further strengthen the manuscript:

1. FRET was used as a proxy for kinase function in some of the discussions. Although this may be the case as expected based on crystal structures, it is not demonstrated, i.e. when the mCit-TwcK-mCFP is stretched, it is unknown whether the kinase regains activity. The current controls for TwcK activity (Figure 1C) only demonstrated the lack of kinase activity for TwcK-FP but not mCit-TwcK-mCFP. There could also be many reasons for the lack of activities, e.g.: (1) NL peptide autoinhibition, (2) misfolding of TwcK domain when tagged by fluorescent proteins, (3) steric-hindrance of target peptide binding by the fluorescent proteins. Can the kinase activity in the presence of force be experimentally demonstrated? I recognize that this may be a very challenging experiment, and outside of the scope of the current manuscript, it should be discussed. In any case, FRET is still valid as a proxy for tension through TwcK. However, arguments based on kinase function during FRET change can only be inferred indirectly. These points should be made more explicitly in the article.

2. Is there any existing estimation on the magnitude of force required to peel the NL domain off from TwcK? I think some estimates of the expected dynamic range of FRET in response to force (order of magnitude estimation) could help interpret the result.

3. Direct comparison of FRET between mCit-TwcK-mCFP (GB282) and mCFP-Ig-mCit (GB284) requires clarification. First, GB282 seems to be expressing the proteins at a higher level comparing to GB284 (Figure 5). Is this true for all worms? Second, it seems that GB284 generally has a lower FRET comparing to GB282. This is puzzling as the authors assumed the Ig domains do not unfold, and hence GB284 should remain at high FRET compared to GB282. Isn't the unfolding of Ig domains a physiological process in muscle function? What is the evidence that Ig domains in the control construct would not unfold? Would it be a better control to remove proteins between the two FRET FPs if this is uncertain? Is maintaining the exact distance between the two FPs crucial?

4. What causes the expression level difference along the length of the worm? It seems that each part of the worm can contract similarly (Figure 4A), which indicates the presence of muscles and sarcomeres along the entire body.

5. The authors have made several arguments regarding intermolecular FRET. The Foster distance for fluorescent proteins is only 4-6 nm. Considering that they sit at a particular location on the long polydomain protein, it would require perfect alignment with nearby twitchins at a very high spatial density for intermolecular FRET to occur, which seems unlikely. Do the authors have evidence that intermolecular FRET is indeed happening?

6. I think it would be beneficial if the authors can show that dorsal and ventral FRET anticorrelate as a validation of their method and strengthen the paper.

7. Figure 1D seems only to suggest that the NL domain is only stretched instead of unbinding then stretched from the kinase. Figure 1A is a clockwise 90-degree turn, which was not clear from the illustration. The ATP pocket should be marked in both figures.

8. Figure 2D: The assessment of distance of 5.3 nm is the distance between the N/C termini of the fluorescent proteins, but not the fluorophores. The two fluorophores' distance and orientation would affect the coupling and FRET between them. Can these values be estimated? In addition, are the observed FRET in vivo similar to FRET observed in vitro using recombinant proteins?

9. Figure 5: The fluorescence intensity is saturated. Is this a display issue, or is the fluorescence data collected saturated? If display issue, I would recommend a non-linear colour legend to display the full dynamic range. If the collection is saturated, then wherever saturation occurs should be excluded for all analyses.

*Reviewer #3* (*Recommendations for the authors*):

1. The authors address the importance of discriminating between the desired intramolecular FRET signal versus any intermolecular FRET, which, given the tight molecular packing of the myofilament lattice, is an important concern to be addressed. Their approach was to introduce the FRET fluorophores into two different positions along the transgenic reporter construct and ultimately, for the test construct, in the gene edited nematodes.

However, for force to act across the kinase region, it would have to be bound to two components of the myofilament lattice moving with respect to each other (explaining sufficient conformational changes to account for the FRET changes and induce an open kinase conformation) by its N- versus C-terminal regions. Therefore, a C-terminally integrated FRET may be in a different molecular and mechanical environment than the regions immediately around the kinase domain.

Ideally, to assess the level of intermolecular FRET, two constructs with Donor or Acceptor fluorophore only, but in their correct positions for the complete sensor, are co-expressed. Any FRET signal detected and any changes in FRET would then be attributable to intermolecular FRET occurring at the site of the sensor due to spatial proximity of the donor and acceptor fluorophores.

2. Figure 5 B and C indicate that expression of the transgenic FRET sensor seems higher in GB282 than in GB284, and this appears to correlate with higher FRET signals. Has the FRET signal been corrected for expression levels or donor-acceptor intensity?

3. Given the major achievement of generating the GB287 (or GB286?) genome-edited nematode line, it is surprising that its characterisation is terse and the results are banished to the supplements. Could FRET videos of this line be shown? Where in the sarcomere does the FRET probe localise?

4. It seems reasonable to postulate that the relationship "between the fluorescence signal and the curvature dynamics … is causal" – but is it due to active force? Would the FRET signal change persist if the worms were paralysed or anaesthetised, e.g. with BDM or a sodium channel blocker, e.g. tricaine? In other words, is this a geometric (lattice packing) or biomechanical (active forces) effect, as far as this can be separated?

5. It is intriguing that the largest decrease in FRET occurs in relaxed sarcomeres, implying that the kinase region would be stretched preferentially when the myosin heads are relaxed. Again, could this hypothesis be tested by interfering with the actomyosin cycle, for example with para-nitroblebbistatin? Could the authors propose, with data from point 3, a hypothetical arrangement of twitchin in the myofilament lattice that would satisfy these observations?

---

## [Author Response]

Reviewer #1 (Recommendations for the authors):Many of the readers will not be familiar with *C. elegans* motility and the manuscript would benefit from having a supplemental video of a moving worm, perhaps captured by each of the imaging modalities.

In response to this excellent suggestion, we have provided a video of a moving worm as a Supplemental Video.

Similarly, there should be some description of what is the period of the undulating waves.

We have added a description in the text (page 8).

It took me a while to wrap my head around the kymographs in Figure 4 and subsequent figures and it would be helpful to have a bit more discussion of how they are generated and interpreted. I assume that each of the diagonals in the kymographs represent a single contractive event. Can you average 10 or more of these in cases where the direction and period are very similar to get cleaner signals?

We agree with the reviewer that the kymographs might be difficult to understand by readers who are not familiar with them. We would like to note that Figure 4 shows specific examples. Because every worm moves in an individualistic way and every cycle is different (in frequency/time), it is very difficult to align the cycles without introducing artifacts. The only way we can confidently say that curvature may be correlated with signals is to do correlation analysis on each incidence, rather than averages of some sort. Therefore, while we find the suggestion to be excellent, on technical grounds we are currently not able to average different cycles without compromising and obscuring the signals.

Is there any way to instantaneously paralyze the worms and freeze them in their undulating shapes? If so, could this simplify the imaging?

We have absolutely thought about ways to slow down the motion of the worms or capture them (i.e. paralyze them) in given mechanical states, but it proved to be experimentally unattainable. For instance, we have put worms on ice, put them in microfluidic devices containing channels with specific shapes, and put them in gels with controlled viscosity. Unfortunately, all of those attempts failed to show interpretable changes of the fluorescence signal for a variety of reasons:

(1) No movement of the animals was observed essentially giving only one data point (very little information),

(2) Very importantly, there was no good control to compare to,

(3) Photobleaching was fast,

(4) Optical artifacts were prominent in the channels of microfluidic devices.

After devoting significant efforts to these attempts and establishing their impracticability, we concluded that the study of worms swimming freely and unhindered was the more practicable and productive approach.

I think that the data shown on the knock-out/knock-in worms is important and I suggest moving it to the main text. Figure 6 could go to the supplement.

In accord with the reviewer’s suggestion, we have

– moved the previous Figure 6 to Supplemental Figure 4

– renamed the original Figure 7 to Figure 6

– created a new Figure 7 showing the results from the CRISPR-generated strain expressing endogenous twitchin with the fluorescent probes on either side of twitchin kinase.

As noted below, we have also added parts A and B demonstrating that this full length twitchin is expressed in muscle, and localizes properly to sarcomeric A-bands.

I do not see where Figure 3E is mentioned in the text.Figure 4D,E. Describe the colors used for the two traces in each panel.

We thank the reviewer for identifying these oversights on our part. Figure 3E is now mentioned (page 10). We have also amended the legend for Figure 4 (D,E) to:

“Sample traces of normalized curvatures (black), and mCFP (blue) in D, and mCitrine (yellow) in E, intensities for a given point along the animal.”

Reviewer #2 (Recommendations for the authors):I enjoyed reading this manuscript and appreciated the transparent discussion of many potential issues. In addition to the issues pointed out in the public review, addressing the following points could further strengthen the manuscript:1. FRET was used as a proxy for kinase function in some of the discussions. Although this may be the case as expected based on crystal structures, it is not demonstrated, i.e. when the mCit-TwcK-mCFP is stretched, it is unknown whether the kinase regains activity. The current controls for TwcK activity (Figure 1C) only demonstrated the lack of kinase activity for TwcK-FP but not mCit-TwcK-mCFP. There could also be many reasons for the lack of activities, e.g.: (1) NL peptide autoinhibition, (2) misfolding of TwcK domain when tagged by fluorescent proteins, (3) steric-hindrance of target peptide binding by the fluorescent proteins. Can the kinase activity in the presence of force be experimentally demonstrated? I recognize that this may be a very challenging experiment, and outside of the scope of the current manuscript, it should be discussed. In any case, FRET is still valid as a proxy for tension through TwcK. However, arguments based on kinase function during FRET change can only be inferred indirectly. These points should be made more explicitly in the article.

During the course of this study, we did not succeed in producing recombinant mCit-NL-TwcK-CRD-mCFP protein in *E. coli* in appropriate yield as to permit performing catalytic assays. Thus, the controls for TwcK activity upon fusion to FP proteins could only be performed with the singly fused species (mCit-NL-TwcK-CRD or NL-TwcK-CRD-mCFP). Regarding the absence of activity in the singly fused variants, scenario (1), i.e. NL peptide autoinhibition (as well as CRD autoinhibition) was indeed the positive and desired outcome that we were looking to validate. In agreement with our characterization of TwcK phosphotransfer activity (in von Castelmmur et al., 2012, PNAS; Williams et al., JMB, 2018), no activity can be measured if NL and CRD tails remain folded onto the kinase. Thus, an absence of catalysis due to NL/CRD inhibition was the expected result. Regarding scenario (2), we consider this to be highly unlikely, given the fact that the samples expressed in high yields, had sharp and well-formed chromatographic profiles in gel exclusion chromatography, were stable upon storage and did not exhibit any degradation/precipitation/aggregation tendency characteristic of misfolded proteins. Similarly, a total ablation of catalysis due to steric-hindrance caused by the unspecific binding of the FP moieties to the target peptide binding site of an otherwise free kinase (3) seems to us difficult to envision. In our experience, TwcK shows very high levels of phosphotransfer activity and even the mutation of a single amino acid in the NL leads to measurable levels of catalysis (Bogomolovas et al., EMBO Rep, 2021). If the packing of the NL or CRD tails would have been disrupted, unspecific binding of the FP proteins (in both N- and C-terminal geometries) would be unlikely to result in such a total and robust blockage of the active site as to abolish catalysis. In our view, the most suitable interpretation of results is scenario (1).

Regarding the demonstration of the catalytic activation of TwcK by stretch, it is indeed the case that this remains unproven. We are not aware of any methodology that permits applying high levels of stretch along a defined molecular axis as to induce tail-unfolding, while also being compatible with the measurement of catalysis. Existing single molecule methodologies are not applicable for this goal. To perform this demonstration in vivo instead would be technically very challenging, especially since the in vivo substrate for TwcK is yet to be identified. Thus, as the referee correctly points out, our inference of phosphotransfer activation based on FRET changes is only indirect. We thank the reviewer for bringing to our attention this important point. We state now more directly to the reader the speculative nature of our reasoning in the Discussion (~pg 20).

2. Is there any existing estimation on the magnitude of force required to peel the NL domain off from TwcK? I think some estimates of the expected dynamic range of FRET in response to force (order of magnitude estimation) could help interpret the result.

To our knowledge, the only available data on the force required to unravel the NL from TwcK derive from the AFM experiments by Greene et al., 2008, Biophys J. At the time of those experiments, the N-terminal sequence of TwcK was not yet known to be a folded fraction of TwcK. This knowledge became available in 2012 (von Castelmur et al., PNAS). Nevertheless, the force response profile of TwcK in the AFM experiments only revealed peaks originating from known 3D-domains, i.e. the kinase lobes and the flanking Fn/Ig domains. Force peaks that could be attributed to the NL or CRD extensions could not be identified. This suggested that the unfolding of these extensions must have occurred at forces below the measurable threshold of the instrumentation in that work, <10pN. However, because of the fast pulling speeds and the single molecule nature of AFM experiments, it is generally believed that forces measured using AFM are not necessarily representative of the in vivo context.

In regards to the dynamic range of FRET, such measurements can, in principle, be an absolute measurement as the reviewer suggests. However, because the experimental system (including the worm, the sensor, and the microscopy system) in this work is highly complex and noisy, such absolute measurements are technically neither feasible nor reliable. Thus, in our study we have correlated the peak and valley of the FRET signal to curvature; this has two advantages – we have thousands of measurements to average out the noise, and we do not rely on the absolute value of the FRET signal. Therefore, a dynamic range for the FRET signal in the sense requested cannot be given.

3. Direct comparison of FRET between mCit-TwcK-mCFP (GB282) and mCFP-Ig-mCit (GB284) requires clarification. First, GB282 seems to be expressing the proteins at a higher level comparing to GB284 (Figure 5). Is this true for all worms? Second, it seems that GB284 generally has a lower FRET comparing to GB282. This is puzzling as the authors assumed the Ig domains do not unfold, and hence GB284 should remain at high FRET compared to GB282. Isn't the unfolding of Ig domains a physiological process in muscle function? What is the evidence that Ig domains in the control construct would not unfold? Would it be a better control to remove proteins between the two FRET FPs if this is uncertain? Is maintaining the exact distance between the two FPs crucial?

Stimulated by the reviewer’s comment, we have performed further experimentation to quantitate expression in both strains GB282 and GB284. Using quantitative Western blots, we have now found that the control strain GB284 expresses 2x higher levels of fusion protein than the test strain GB282 (with N=4, mean = 2.10). This information is provided in our revised manuscript (Supp Figure 2). The level of expression, however, is not directly responsible for the FRET result. In this respect, it is important to note that the values reported and mutually compared in our study are FRET to curvature correlations, and not absolute values. It is therefore to be expected that the ΔFRET signal from the control strain, GB284, should indeed be lower if Ig28 is a non-extensible control element, as postulated. In this regard, it should be noted that the force response of Ig28 has been previously measured using AFM (Greene et al., 2008, Biophys J). The authors studied a poly-Ig segment corresponding to domains Ig26-Ig30 from twitchin, where domains displayed a mean unfolding force of 93±25 pN. This indicated that the domains are mechanically stable. Regarding the potential unfolding of Ig domains during sarcomere function, this has been best investigated in the vertebrate protein titin, where nevertheless the question remains highly debated. It is believed that in healthy muscle only a limited number of Ig domains may unfold in the stretched sarcomere, Ig unfolding likely being a stochastic event and, thereby, probably a rare event instead of a primary molecular mechanism. Based on available knowledge, it is reasonable to assume that Ig28 in the control strain will remain folded in *C. elegans* in the large majority of molecular copies of twitchin in the sarcomere. This also agrees with the low ΔFRET signal for this strain. To better clarify these points to the reader, we have added explanatory text to pg 8 and 9 of the revised manuscript.

4. What causes the expression level difference along the length of the worm? It seems that each part of the worm can contract similarly (Figure 4A), which indicates the presence of muscles and sarcomeres along the entire body.

The apparent difference in fluorescence signal from front to back of the animal likely reflects changes in the size of the body wall muscle cells; they are widest and have the most numbers of sarcomeres in the middle of the animal compared with the anterior and posterior ends.

5. The authors have made several arguments regarding intermolecular FRET. The Foster distance for fluorescent proteins is only 4-6 nm. Considering that they sit at a particular location on the long polydomain protein, it would require perfect alignment with nearby twitchins at a very high spatial density for intermolecular FRET to occur, which seems unlikely. Do the authors have evidence that intermolecular FRET is indeed happening?

The reviewer makes an excellent point. We do not have any experimental evidence of intermolecular FRET taking place in the sarcomere and, if so, of how strong this contribution is; we only suggest that it could happen. As FRET transfer still takes place beyond the Förster distance (up to ca 10 nm), we find it conceivable that in the crowded environment of the sarcomere FRET happens to some extent across molecules. Thus, we speculate on its occurrence as one of the various potential sources of noise in the data. In the revised manuscript, we state now more clearly to the reader the speculative nature of our suggestion (pg 15, 17 and 18).

6. I think it would be beneficial if the authors can show that dorsal and ventral FRET anticorrelate as a validation of their method and strengthen the paper.

We thank the reviewer for making us aware of the need for clarification. The data already show this – we correlate the FRET to the curvature (positive/negative curvatures would indicate whether contraction and extension is happening). We did not track ventral or dorsal because that would require identifying where the vulva is, and it is unnecessary because being dorsal or ventral does not change the fact that the sarcomere can contract or extend – both sides do. What matters is where in the contraction/extension cycle the muscle is in (for which curvature is a good proxy), correlated to the FRET signal. This is what was plotted.

7. Figure 1D seems only to suggest that the NL domain is only stretched instead of unbinding then stretched from the kinase. Figure 1A is a clockwise 90-degree turn, which was not clear from the illustration. The ATP pocket should be marked in both figures.

Figure 1D is meant to provide a simple, schematic overview of the intended functional principle of the FRET sensor. The non-stretched state is meant to represent the basal molecular state, where the NL is bound to the kinase. Unbinding and extension of the NL happen concurrently upon stretch and are not separated steps. We have amended the figure (left panel) to represent this better as well as we now state this explicitly in the figure legend. In Figure 1A, left and right panels are related by a 90^o^ rotation around an axis contained within the plane of the paper, as correctly identified by the reviewer. We have annotated with “90^o^” the rotation symbol in the figure to clarify this. The ATP pocket is marked in Figure 1A. Attempts to mark the ATP pocket also in Figure 1D, as requested, have only resulted in largely unsatisfactory representations that obscure the main graphical elements of that figure and, thereby, its primary message on the function of the FRET sensor. Thus, we kindly request that the schematic Figure 1D is considered as provided.

8. Figure 2D: The assessment of distance of 5.3 nm is the distance between the N/C termini of the fluorescent proteins, but not the fluorophores. The two fluorophores' distance and orientation would affect the coupling and FRET between them. Can these values be estimated? In addition, are the observed FRET in vivo similar to FRET observed in vitro using recombinant proteins?

In FP proteins, the fluorophore is contained within the protein core, encased within the β-barrel fold of the protein. Based on crystal structures of the FP proteins employed in this study, the fluorophore is known to be located approx. 2.4 nm from the terminal pole of the FP fold. FRET, however, will not occur between the fluorophore and the termini, but between the fluorophores across the two neighbouring FP units. It is indeed likely that the attachment sites at the termini act as pivot points that permit oscillatory motions of the FP subunits around an average position. However, we do not know what the exact orientation of the FP subunit respect to each other in our constructs is or the extent of their respective dynamics. Obtaining this information is an extremely demanding exercise in itself. Thus, we cannot provide here realistic quantitative estimates. As a rough estimate and assuming a cone of free motion around pivot points (i.e. attachment points), we could predict a speculative fluorophore distance range of 2.9 – 7.9 nm (i.e. 5.3 nm +/- 2.4 nm). These distances would roughly correspond, respectively, to a minimal distance where the two FPs would directly co-localize laterally and to a maximal coaxial opposition of the FPs separated by the kinase domain in between. However, it is important to note that FRET changes in this study were obtained by averaging thousands of measurements, which would remove non-systematic variability in the observations, such as precisely FRET differences resulting from dynamic positioning and reorientation of fluorophores in individual molecules. This is stated in pg 13 of the manuscript.

Regarding the comparison of FRET in vitro and in vivo, this is not within our reach. On one hand, we have not succeeded in producing a recombinant mCit-NL-TwcK-CRD-mCFP sample in appropriate quantity as to permit experimentation; more importantly, we are not aware of any current technology that would allow us to measure ΔFRET values in vitro in function of physiological-levels of stretch in such recombinant samples.

9. Figure 5: The fluorescence intensity is saturated. Is this a display issue, or is the fluorescence data collected saturated? If display issue, I would recommend a non-linear colour legend to display the full dynamic range. If the collection is saturated, then wherever saturation occurs should be excluded for all analyses.

This is a choice of display that, in our opinion, shows with best clarity the contrast between high and low values in curvature cycles.

Reviewer #3 (Recommendations for the authors):1. The authors address the importance of discriminating between the desired intramolecular FRET signal versus any intermolecular FRET, which, given the tight molecular packing of the myofilament lattice, is an important concern to be addressed. Their approach was to introduce the FRET fluorophores into two different positions along the transgenic reporter construct and ultimately, for the test construct, in the gene edited nematodes.However, for force to act across the kinase region, it would have to be bound to two components of the myofilament lattice moving with respect to each other (explaining sufficient conformational changes to account for the FRET changes and induce an open kinase conformation) by its N- versus C-terminal regions. Therefore, a C-terminally integrated FRET may be in a different molecular and mechanical environment than the regions immediately around the kinase domain.

The reviewer makes an excellent point that in order for pulling to be “sensed” by the kinase region, segments N- and C-terminal of it must be anchored. Unfortunately, the exact structure of the twitchin filament and its integration in the sarcomeric lattice is currently unknown. It is also not known what the magnitude of the shearing forces resulting from hydrodynamic effects in a sliding sarcomere lattice are. Thus, we cannot assign at this time an architectural cause to the changes observed. It should be noted that a similar stretch-unfolding mechanism was first proposed for titin kinase (Lange et al., Science, 2005; Puchner et al., PNAS, 2008). The A-band fraction of the titin filament is known to be integrated within the thick filament, while the very C-terminus of titin interacts with M-line components, accounting in this way for the N- and C-terminal anchoring of the kinase. The sarcomeric context of twitchin is largely unstudied and equivalent interactions have not been revealed to this date, so that a structural rational for the changes observed cannot be provided at the moment (and, we do know that twtichin does not extend into the M-line). A speculation, however, could be made based on the known facts that: (1) the main portion of twitchin consists of multiple copies of Fn-Fn-Ig similar to the main myosin binding segments of human titin’s A-band (Kontrogianni-Konstantopoulos A et al., 2009); (2) at least in vitro, the substrate for Aplysia TwcK, is regulatory myosin light chain (Heierhorst et al., 1995); and (3) evidence that twitchin in molluscs acts as a bridge between thick and thin filaments and facilitates the “catch state” (Butler and Siegman, 2010). Building on these points, we envision the following hypothetical arrangement of twitchin in the sarcomere: the vast majority of the protein, from the N-terminus to just before the kinase domain is associated with the shaft of the thick filament; the kinase domain is associated with the myosin head; and the C-terminal tandem Ig domains are associated with the thin filament. With this arrangement, because most of twitchin is anchored to the thick filament, and the C-terminus is anchored to the thin filament, when myosin heads move away from the thin filaments during muscle relaxation, force is applied to move the NL or the CRD from the kinase catalytic core, and perhaps resulting in phosphotransferase activity. However, given the highly speculative nature of this proposal and out of caution, we prefer not to include this potential arrangement mode in our manuscript.

Ideally, to assess the level of intermolecular FRET, two constructs with Donor or Acceptor fluorophore only, but in their correct positions for the complete sensor, are co-expressed. Any FRET signal detected and any changes in FRET would then be attributable to intermolecular FRET occurring at the site of the sensor due to spatial proximity of the donor and acceptor fluorophores.

We agree with the reviewer that, had we made such strains, they would have provided information on the intermolecular FRET signal. However, it is important to note that for such an experiment to yield the desired information, the worm strain would need to co-express the twitchin fragment variants at comparable expression levels and the two variants would need to co-integrate themselves in the sarcomere also at comparable ratios and homogeneously, which would require validation. Overall, we feel that the technical complexity of such an experiment would be high. It must also be noted that the expression of the fluorophores in any independently generated strain will be variable, and, thus, measurements could not be directly compared to those from other strains, so that it could not serve as a direct reference for the FRET strain in this work. In our view, such an experiment is not necessary to draw the conclusion that we draw. Important in this work is that, if intermolecular FRET is taking place, its contribution to the signal can be expected to be comparable in control and test strains. Also important to note is that we draw our conclusions mainly based on the comparisons of the correlations of the FRET and curvature dynamics. Finally, as stated by reviewer #2 in their comment, it is worth noticing that FRET only occurs at rather short distances, a condition that might be less well met by neighboring molecules. In brief, we feel that although intermolecular FRET must be considered, measuring its contribution experimentally is challenging and that, by calculating comparative ΔFRET values, we have ensured that its contribution is unlikely to be such that it invalidates the results.

2. Figure 5 B and C indicate that expression of the transgenic FRET sensor seems higher in GB282 than in GB284, and this appears to correlate with higher FRET signals. Has the FRET signal been corrected for expression levels or donor-acceptor intensity?

As discussed above in response to Reviewer 2’s comments, the intensity shown in Figure 5 is adjusted to show the peaks and valleys, and do not correlate to the absolute signal. Further, the expression of the fluorophores are not uniform throughout the tissue, and this is something that we cannot control. Regarding levels of expression, as requested, we have performed a quantitative western blot analysis to determine the protein expression levels from the transgenes in GB282 and GB284 [Supp Figure 2]. We show in fact that the expression level from GB284, the control line, is actually 2.1 fold higher than from GB282, the tester line, so the higher fluorescent signal from GB282 is indeed not because of a higher expression level. However, the exact levels of expression do not directly affect the results. This is because the donor and the acceptor are at 1:1 ratio in all cells by the design of the sensor and we apply ratiometric FRET calculations that make the absolute levels of the FRET intensity irrelevant.

3. Given the major achievement of generating the GB287 (or GB286?) genome-edited nematode line, it is surprising that its characterisation is terse and the results are banished to the supplements. Could FRET videos of this line be shown? Where in the sarcomere does the FRET probe localise?

We thank the reviewer for recognizing how difficult it was to create this genome-edited line, which is called GB287. In agreement with the reviewer, we have moved the previous supplemental Figure 4 to a regular Figure 7. In answer to the reviewer’s question, we have added two parts to this figure, A and B, showing that the tagged full-length twitchin is expressed in the appropriate muscle cells (part A), and showing that it is properly localized to the A-bands of the sarcomere. The strain showed normal locomotion and there was nothing remarkable about this strain in particular. We state now these results in pg 17 of the revised manuscript. However, the fluorophores were very dim since the protein is expressed from its own single-copy gene as opposed to overexpression from transgenic arrays that carry multiple copies. We therefore have decided just to show the processed data.

4. It seems reasonable to postulate that the relationship "between the fluorescence signal and the curvature dynamics … is causal" – but is it due to active force? Would the FRET signal change persist if the worms were paralysed or anaesthetised, e.g. with BDM or a sodium channel blocker, e.g. tricaine? In other words, is this a geometric (lattice packing) or biomechanical (active forces) effect, as far as this can be separated?

Please see responses to Review 1 (4th comment).

5. It is intriguing that the largest decrease in FRET occurs in relaxed sarcomeres, implying that the kinase region would be stretched preferentially when the myosin heads are relaxed. Again, could this hypothesis be tested by interfering with the actomyosin cycle, for example with para-nitroblebbistatin? Could the authors propose, with data from point 3, a hypothetical arrangement of twitchin in the myofilament lattice that would satisfy these observations?

We thank the reviewer for this idea. In fact, we did also try using pharmacological methods to test the hypothesis; for instance, we used levamisole, an acetyl choline receptor agonist to induce hypercontracted paralysis. Again similar to putting worms on ice or in channels, drugged worms did not produce motion and only gave us an equivalent of a single data point; the animals photobleached easily, so the experiment did not produce useful results.

Identifying the exact mechanistic sources of stretch and correlating this to the mechanical states of the acto-myosin motors is certainly a worthy line of future experimentation.